# SleepSMC: Ubiquitous Sleep Staging via Supervised Multimodal Coordination

**Shuo Ma**[1,2], **Yingwei Zhang**[1,2], **Yiqiang Chen**[1,2,3],[*] **Hualei Wang**[1,2], **Yuan Jin**[4],
**Wei Zhang**[4], **Ziyu Jia**[5,6]

[1] Beijing Key Lab. of Mobile Computing and Pervasive Device, Institute of Computing Technology, Chinese Academy of Sciences [2] University of Chinese Academy of Sciences [3] Pengcheng Laboratory

[4] AI Dream (Zhuhai) Intelligent Technology Co. Ltd. [5] Beijing Key Laboratory of Brainnetome and Brain-Computer Interface, Institute of Automation, Chinese Academy of Sciences

[6] Brainnetome Center, Institute of Automation, Chinese Academy of Sciences

{mashuo20g, zhangyingwei, yqchen, wanghualei23s}@ict.ac.cn,
{john, david.zhang}@aidream.cn,jia.ziyu@outlook.com

## Abstract

Sleep staging is critical for assessing sleep quality and tracking health. Polysomnography (PSG) provides comprehensive multimodal sleep-related information, but its complexity and impracticality limit its practical use in daily and ubiquitous monitoring. Conversely, unimodal devices offer more convenience but less accuracy. Existing multimodal learning paradigms typically assume that the data types remain consistent between the training and testing phases. This makes it challenging to leverage information from other modalities in ubiquitous scenarios (e.g., at home) where only one modality is available. To address this issue, we introduce a novel framework for ubiquitous **Sleep** staging via **S**upervised **M**ultimodal **C**oordination, called **SleepSMC**. To capture category-related consistency and complementarity across modality-level instances, we propose supervised modality-level instance contrastive coordination. Specifically, modality-level instances within the same category are considered positive pairs, while those from different categories are considered negative pairs. To explore the varying reliability of auxiliary modalities, we calculate uncertainty estimates based on the variance in confidence scores for correct predictions during multiple rounds of random masks. These uncertainty estimates are employed to assign adaptive weights to multiple auxiliary modalities during contrastive learning, ensuring that the primary modality learns from high-quality, category-related features. Experimental results on four public datasets, ISRUC-S3, MASS-SS3, Sleep-EDF-78, and ISRUC-S1, show that SleepSMC achieves state-of-the-art cross-subject performance. SleepSMC significantly improves performance when only one modality is present during testing, making it suitable for ubiquitous sleep monitoring.

## 1 Introduction

Sleep staging(Liu & Jia, 2023; Tu et al., 2016) is crucial for health monitoring, sleep quality assessment, and the diagnosis of neurological disorders (Zhang et al., 2019; Jia et al., 2022; Scott et al., 2022). According to the American Academy of Sleep Medicine (AASM) (Berry et al., 2012) standards, sleep is categorized into five stages. Sleep staging relies on the integrated representation of various modalities within multimodal polysomnography (PSG) (Tăutan et al., 2020; Kwon et al., 2021), each containing shared and unique sleep-related information. Researchers usually analyze the performance of these signals under the guidance of the AASM to determine sleep stages.

In addition to PSG, sleep staging can also be achieved using multi-channel electroencephalography (EEG) (Liu & Jia, 2023), single-channel EEG (Phan et al., 2022), wearable devices like ear-EEG (Mikkelsen et al., 2019) and smartwatch (Chang et al., 2018), or even more comfortable contactless devices like Ballistocardiography (BCG) pressure sensors (Mengxing et al., 2019) and

---

[*]Corresponding author: Yiqiang Chen

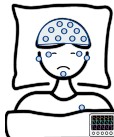
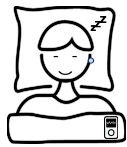

(a) Uncomfortable multimodal PSG    (b) Comfortable unimodal device

Figure 1: Multimodal devices are accurate but uncomfortable, while unimodal devices are less accurate but more comfortable, making them ideal for use in ubiquitous scenarios (e.g., at home).

Radar-based systems (Walid et al., 2021). However, PSG and multi-channel EEG, due to their rich brain-related information, provide the highest accuracy for sleep staging. Single-channel EEG and wearable devices, which only capture partial or indirect brain signals, generally offer lower accuracy, while contactless devices tend to have the lowest precision. As shown in Figure 1, while complex multimodal devices enhance accuracy, they also significantly impact sleep comfort, making them less suitable for ubiquitous scenarios (e.g., at home) (Ma et al., 2024a). Leveraging multimodal data to improve not only multimodal testing performance but also unimodal performance remains a key challenge.

However, most multimodal learning approaches (Jia et al., 2020; 2021c; Yubo et al., 2022; Cao et al., 2024; Ma et al., 2023; 2024b) assume that consistent modalities available during training and testing are consistent, leading it difficult to maintain accuracy when only one modality is available during testing. The **multimodal coordination** is dedicated to addressing this issue. It employs methods like feature alignment to explore modality-level consistency and complementarity during coordinated training, enhancing the performance of multimodal systems and even improving the effectiveness of the single primary modality during testing. For example, Liu et al. (2023b) aligned multimodal representations with EEG representations, distilling the knowledge from multiple modalities into EEG-based models. Liu et al. (2024) employed contrastive learning to align representations both within the ECG modality and between it and the medical text modality, thereby enhancing the performance during testing when only ECG is available. However, existing methods lack targeted supervision (Khosla et al., 2020; Mai et al., 2023) during knowledge transfer, making it difficult to learn category-specific features effectively, and the information from auxiliary modalities is not effectively filtered.

To address these challenges, we propose SleepSMC, a ubiquitous sleep staging method that combines supervised modality-level instance contrastive coordination and uncertainty-based feature weighting. SleepSMC leverages contrastive coordination to effectively align category-related information across modality-level instances. By bringing same-category instances closer (including alignment within the same modality) and pushing different-category instances apart, SleepSMC ensures the model learns discriminative features that incorporate both consistency and complementarity. To enhance this learning process, SleepSMC introduces uncertainty-based feature weighting for auxiliary modalities. It estimates the robustness of each auxiliary modality under random masks and adjusts their contribution accordingly in contrastive learning, ensuring that it emphasizes more reliable and category-related features. During testing, SleepSMC demonstrates superior performance not only when multiple modalities are available, but also in unimodal scenarios, bridging the gap between multimodal training and real-world deployment.

To summarize, our contributions are as follows:

- To the best of our knowledge, we first introduce multimodal collaboration in sleep staging, leveraging multiple auxiliary modalities to improve the performance of primary modality-based sleep staging in an end-to-end manner.

- We utilize supervised modality-level instance contrastive coordination to capture category-related consistency and complementarity across intra-modality and inter-modality.

- We utilize uncertainty estimates to adaptively weight auxiliary modality features during training. This approach ensures that more reliable auxiliary modality features contribute more significantly to the contrastive learning process, thereby improving the performance of the primary modality.

- Extensive experimental results on four public datasets demonstrate that SleepSMC not only enhances multimodal across-subject sleep staging performance but also significantly boosts unimodal across-subject sleep staging performance.

## 2 RELATED WORK

**Sleep Staging.** Sleep staging is crucial for monitoring sleep quality and diagnosing neurological conditions. According to the AASM standard, sleep comprises five stages: Wake, NREM stages N1, N2, N3, and REM. Sleep staging is typically based on deep learning models, like CNN (Tsinalis et al., 2016; Supratak et al., 2017), RNN (Bresch et al., 2018; Supratak & Guo, 2020), and LSTM (Liang et al., 2023a; Phyo et al., 2022), which enables the automatic extraction of spatial and temporal features from PSG signals. To exploit complementarity and consistency across modalities in PSG, researchers have designed modality-specific feature extraction and fusion modules(Xiang et al., 2023; Yubo et al., 2022). However, the reliance on multiple modalities is impractical in ubiquitous settings where only one modality is available. This limitation underscores the need for models to leverage multimodal data during training but maintain high performance with only one modality during testing. To solve this issue, Mikkelsen et al. (2019) developed unimodal portable wearable devices, ear EEG to enhance the comfort of sleep monitoring in ubiquitous scenarios. Jia et al. (2024) and Liang et al. (2023a) transferred knowledge from large-scale models to lightweight models through multi-level alignment for wearable device use. However, due to the limited local information that unimodal devices can perceive, the accuracy of unimodal sleep staging is constrained.

**Multimodal Coordination.** The limitations of low accuracy in unimodal sleep staging highlight the need for multimodal coordination methods (Zadeh et al., 2020; Liang et al., 2023b; Rahate et al., 2022). Multimodal coordination methods (Jia et al., 2021a; Jiang et al., 2024; Lin & Hu, 2024; Liu et al., 2023a) leverage multimodal data to enhance the performance of a primary modality during training, while relying on a single modality during testing, making them better suited for real-world applications. For instance, Liu et al. (2024) employed unsupervised contrastive learning to align ECG data with medical text, improving ECG-only scenarios. Despite its successes, this approach is complex, involving pre-training and fine-tuning. Liu et al. (2023b) leveraged the dual-modal features of EEG and skin response, along with unimodal EEG, for knowledge distillation to boost EEG-based emotion recognition. The above methods all rely solely on the consistency of multimodal data itself and lack the learning of category-related information.

**Uncertainty Estimation.** Most uncertainty estimation methods focus solely on input with only a single modality, exploring robustness evaluation (Salman et al., 2019; Rosenfeld et al., 2020; Xu et al., 2020; Lyu et al., 2021) and decision boundary adjustments (Weng et al., 2018; Leino et al., 2021) for unimodal models. Existing Multimodal Uncertainty Estimation methods (Yang et al., 2024; Tellamekala et al., 2023) typically play a role in multimodal fusion. For instance, Tellamekala et al. (2023) quantified uncertainty in modalities and employed calibration and ordinal constraints to multimodal models, improving emotion recognition performance and robustness. These works are tailored for effective multimodal fusion, including retaining useful information from each modality while removing redundant or irrelevant data. They are always requiring all the modalities present during training to be available during testing. If only a single modality is used during testing, performance suffers severely, failing unimodal applications.

## 3 PRELIMINARY

The training set is $\mathcal{D}_{train} = \{(x_i, y_i) | i \in \{1, \ldots, I\}\}$ and the testing set is $\mathcal{D}_{test} = \{x_j | j \in \{1, \ldots, J\}\}$. The training data $\mathcal{D}_{train}$ and testing data $\mathcal{D}_{test}$ are from different subjects. $x_i = [x_i^e, x_i^o, x_i^m]$ and $x_j = [x_j^e, x_j^o, x_j^m]$ are PSG signals containing synchronized EEG, EOG, EMG, and ECG, and each epoch lasts 30 seconds. $x_i$ and $x_j$ are the $i$-th and $j$-th samples of training and testing set, respectively. $y_i \in \{1, 2, \ldots, K\}$ is the corresponding sleep category and $K$ is the total number of categories. $I$ and $J$ are the training and testing set sample numbers, respectively. $U$ is the number of modalities, and $U \times I$ is the number of all modality-level instances.

Sleep staging problem is defined as $\hat{y}_i = \text{argmax}_k [G_y(G_f(x_i))]_k$, where $\hat{y}_i$ denotes the predicted category and $k \in \{1, 2, ..., K\}$ denotes the index of category. $G_f = [G_f^e, G_f^o, G_f^m]$ denote modality-specific feature extractors for EEG, EOG, chin EMG, respectively, and the parameter of $G_f$ denoted

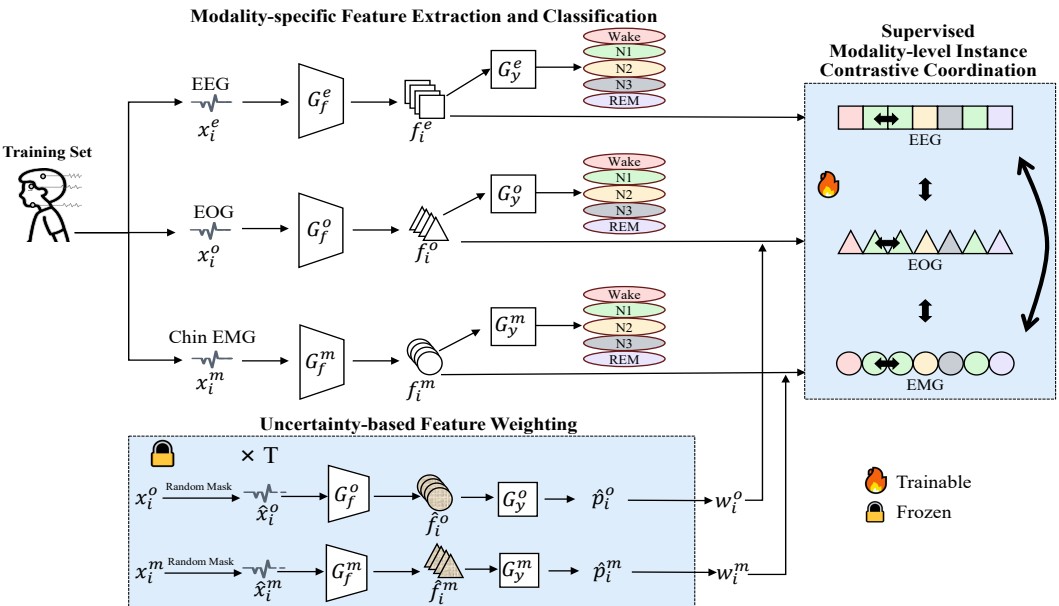

Figure 2: The pipeline of SleepSMC begins with modality-specific feature extraction and classification for each input modality. Subsequently, uncertainty-based feature weighting is employed by estimating the reliability of auxiliary modalities through multiple rounds of random masks. These weights are employed to refine the extracted features during supervised modality-level instance contrastive coordination, which aligns category-related information across modality-level instances. In the figure, EEG is shown as an example of the primary modality. Shapes indicate modalities, colors indicate categories, and bidirectional arrows show the alignment of some positive instance pairs.

as $\theta^f = \left[\theta^{f,e}, \theta^{f,o}, \theta^{f,m}\right]$, $G_y^e$, $G_y^o$, and $G_y^m$ denote modality-specific label classifiers (only one classifier is employed for testing in unimodal testing scenario). $f_i^e$, $f_i^o$ and $f_i^m$ represent the features extracted from synchronous modality-level instances $x_i^e$, $x_i^o$ and $x_i^m$, respectively.

To simplify the symbol expression, we use $u \in \{e, o, m\}$ to denote the modality-related symbol. In this paper, **ubiquitous sleep staging** refers to using only a single primary modality during testing (e.g., EEG), which is more convenient for real-world applications. The **primary modality** is denoted as $u_p \in \{e, o, m\}$, and **auxiliary modalities** are denoted as $u_a \in \{e, o, m\} \setminus u_p$. The auxiliary modalities assist during training but are not used during testing. $\hat{p}_i^u \in \{\hat{p}_i^e, \hat{p}_i^o, \hat{p}_i^m\}$ denote the predicted logits for different modalities from the same $x_i$, while $\hat{y}_i^u \in \{\hat{y}_i^e, \hat{y}_i^o, \hat{y}_i^m\}$ denote the predicted category labels. $f_i^u$ denotes the modality-level feature. The main mathematical notations used in this paper are summarized in Table 4.

## 4 THE PROPOSED SLEEPSMC

The pipeline of SleepSMC is shown in Figure 2, and the corresponding algorithm pseudocode is shown in Appendix A.2. SleepSMC is built upon three key components: 1) Modality-specific feature extraction and classification: This serves as the backbone of SleepSMC, preserving the unique characteristics of each modality. 2) Uncertainty-based feature weighting: This component ensures that more reliable and valuable features contribute more during training. 3) Supervised modality-level instance contrastive coordination: This enables the learning of category-related shared and complementary features across modality-level instances, ensuring effective multimodal coordination for unimodal testing scenarios.

### 4.1 MODALITY-SPECIFIC FEATURE EXTRACTION AND CLASSIFICATION

In SleepSMC, we utilize the FeatureNet (Jia et al., 2021b) (a dual-scale CNN model) as the backbone for modality-specific feature extraction and classification. For a given input sample $x_i$, the modality-

specific feature extractor $G_f^u$ is employed to extract feature $f_i^u$ for the instance $x_i^u$. These features are subsequently employed to predict the sleep stage classification for each modality.

The modality-specific classification loss $\mathcal{L}_{cls}^u$ for modality $u$ is computed as follows:

$$\mathcal{L}_{cls}^u = \frac{1}{I} \sum_{i=1}^{I} \mathcal{L}_{CE}(G_y^u(f_i^u), y_i) \tag{1}$$

where $\mathcal{L}_{CE}$ denotes the cross-entropy loss for the classification task, $G_y^u(f_i^u)$ denotes the predicted category logits for $f_i^u$, $y_i$ denotes the true sleep stage label for sample $i$.

This modality-specific architecture enables the model to effectively preserve the unique information contributed by EEG, EOG, and chin EMG signals.

## 4.2 Uncertainty-Based Feature Weighting

To handle the uncertainty differences between auxiliary modalities, we introduce an uncertainty estimation mechanism that assesses and adaptively weights auxiliary modality features during contrastive learning. This approach allows the model to emphasize reliable auxiliary modality features more while reducing the impact of those with higher uncertainty. Notably, the primary modality is excluded from the uncertainty estimation process and remains unweighted.

**Uncertainty Estimation with Frozen Gradients.** For each auxiliary modality $u_a$, multiple rounds of random masks are employed during training. Specifically, random intervals selected based on a normal distribution are replaced with zero in the input signal $x_i^{u_a}$ for each auxiliary modality. This process is formalized as:

$$\hat{x}_i^{u_a} = \varphi(x_i^{u_a}, \eta) \tag{2}$$

where $\varphi$ represents random mask transformation, and $\eta$ is the percentage of the signal to be replaced with zeros. This transformation is employed $T$ times, generating $T$ different perturbed versions of the signal:

$$\hat{X}_i^{u_a} = \{\hat{x}_{i,1}^{u_a}, \hat{x}_{i,2}^{u_a}, \ldots, \hat{x}_{i,T}^{u_a}\} \tag{3}$$

For each perturbed instance $\hat{x}_{i,t}^{u_a}$, we compute the confidence score $\hat{p}_{i,t}^{u_a}$ for the correct category from modality-specific classifier $G_y^{u_a}$:

$$\hat{p}_{i,t}^{u_a} = \left[ G_y^{u_a}(G_f^{u_a}(\hat{x}_{i,t}^{u_a})) \right]_{y_i} \tag{4}$$

where $y_i$ denotes the ground truth label, and $t$ denotes the transformation index for the same instance.

Then, the uncertainty variance for auxiliary modality $u_a$ is computed as:

$$r_i^{u_a} = \frac{1}{T} \sum_{t=1}^{T} \left( \hat{p}_{i,t}^{u_a} - \frac{1}{T} \sum_{t=1}^{T} \hat{p}_{i,t}^{u_a} \right)^2 \tag{5}$$

Notably, during the entire calculation of uncertainty metrics, all gradients are frozen to ensure that the uncertainty estimation does not interfere with the training process.

**Feature Weighting Based on Uncertainty Estimates.** After calculating the uncertainty variance $r_i^{u_a}$ with frozen gradients, this variance is utilized to weight the auxiliary modality features. This weighting is employed solely for the auxiliary modalities during the contrastive learning process, leaving the primary modality unaffected by the uncertainty estimation.

The feature weighting process involves two key stages. First, an exponential function is employed to the negative value of $-r_i^{u_a}$ for the auxiliary modalities. The transformation ensures that modalities with high uncertainty (large $r_i^{u_a}$) are assigned exponentially smaller weights, while those with low

uncertainty retain larger weights (small $r_i^{u_a}$). The exponential function introduces a smooth, continuous inverse scaling, where weights decrease more sharply for high uncertainty and more gradually for low uncertainty, reflecting the varying tolerance of the model. This process is formalized as:

$$z_i^{u_a} = \exp(-r_i^{u_a}) \tag{6}$$

where $r_i^{u_a}$ is the uncertainty variance for instance $x_i^a$ (i.e., sample $x_i$ with modality $u_a$).

Next, we apply a softmax function to normalize these transformed values between auxiliary modalities and multiply the normalized weights by the number of auxiliary modalities $(U - 1)$ to ensure that their combined contribution is properly scaled. The feature weights for each auxiliary modality $u_a$ are computed as:

$$w_i^{u_a} = \frac{\exp(z_i^{u_a}/\tau)}{\sum_v \exp(z_i^v/\tau)} \odot (U - 1) \tag{7}$$

where $\tau$ denotes the temperature scaling parameter, $v \in \{1, 2, U - 1\}$ indexes all auxiliary modalities. $\odot$ denotes element-wise product.

These weighted modality features are then fed into the contrastive learning process, where gradients are backpropagated to optimize the model. By prioritizing the more reliable features, this method enhances the model's overall robustness and improves its ability to align and coordinate features across modality-level instances. Notably, this weighting mechanism only directly affects contrastive learning loss and does not directly impact classification loss. The weighted features for the auxiliary modalities are computed as follows:

$$f_i^{u_a'} = w_i^{u_a} \odot f_i^{u_a} \tag{8}$$

where $f_i^{u_a}$ represents the features from of auxiliary modality $u_a$, $f_i^{u_a'}$ is the weighted feature for auxiliary modality $u_a$, $\odot$ denotes element-wise product.

**Proofs in Appendix A.5.** We quantify the information transfer using perturbation radius and provide upper and lower bounds to explain how our uncertainty-based weighting works. Additionally, we employ minimax theory to prove that even in the worst-case scenario, where auxiliary modalities have high uncertainty (i.e., low weights), effective information transfer can still be achieved.

4.3 SUPERVISED MODALITY-LEVEL INSTANCE CONTRASTIVE COORDINATION

After modality-specific features are extracted and weighted based on uncertainty, SleepSMC leverages supervised modality-level instance contrastive learning to align modality-level instances of the same sleep stage, while simultaneously separating those of different sleep stages. Modality-level alignment effectively encourages multimodal coordination, particularly in unimodal testing scenarios.

**Contrastive Loss Function.** Firstly, the features from the modality-specific feature extractor be $L_2$-normalized as $\widetilde{f}_i^u = \left\{ \frac{f_i^{u_a'}}{\|f_i^{u_a'}\|}, \frac{f_i^{u_p}}{\|f_i^{u_p}\|} \right\}$. Then, for input data, the supervised contrastive loss is computed by comparing each anchor instance with positive pairs (i.e., instances of the same category) and negative pairs (i.e., instances from different categories), excluding the anchor itself. The supervised contrastive loss is defined as follows:

$$\mathcal{L}_{\text{Con}} = \frac{1}{U \times I} \sum_{i'=1}^{U \times I} \frac{1}{|Q(i')|} \sum_{q \in Q(i')} - \log \frac{\exp(\widetilde{f}_{i'}^u \cdot \widetilde{f}_q^u / \tau)}{\sum_{v'=1}^{U \times I-1} \exp(\widetilde{f}_{i'}^u \cdot \widetilde{f}_{v'}^u / \tau)} \tag{9}$$

where $i' \in \{1, 2, ..., U \times I\}$ indexes of all modality-level instances, $\widetilde{f}_{i'}^u$ and $\widetilde{f}_q^u$ are the normalized weighted features for the anchor instance $x_{i'}^u$ and the positive instance $x_q^u$, respectively. The dot product $\cdot$ measures the similarity between features. $\tau$ denotes the temperature scaling parameter, controlling the sharpness of the similarity scores. $Q(i')$ denotes the set of indices of all positive instances (i.e., those belonging to the same category) relative to the anchor instance. The denominator

sums over all instances, excluding the anchor instance itself, resulting in $(U \times I - 1)$ comparisons, where $v' \in \{1, 2, ..., U \times I - 1\}$ denotes the corresponding index.

This loss function encourages the model to draw instances of the same category closer together within the feature space, while simultaneously pushing instances from different categories further apart. This modality-level alignment captures both inter-modal and intra-modal consistency and complementarity, enabling the model to focus more effectively on category-related information.

**Final Loss and Optimization.** The final objective function combines the supervised contrastive loss with a modality-specific classification loss:

$$\mathcal{L}_{\text{total}} = \sum_{u \in \{e,o,m\}} \mathcal{L}_{\text{cls}}^u + \mathcal{L}_{\text{Con}} \tag{10}$$

where $\mathcal{L}_{\text{cls}}^u$ is the classification loss for all instances with modality $u$.

Finally, the total loss is backpropagated to jointly optimize the model parameters. This joint optimization process enhances the model's performance in cross-subject sleep staging, particularly in unimodal testing scenarios.

## 5 EXPERIMENTS

All experiments are implemented with Python 3.8.5 and Pytorch 1.7.1. We conduct them on a computer server with 640GB RAM and two NVIDIA RTX A5000 GPUs with 24GB VRAM each. Detailed experiment settings are shown in Appendix A.3.

### 5.1 DATASET AND DATA PROCESSING

We evaluate SleepSMC on four public datasets: ISRUC-S3 (Khalighi et al., 2016), MASS-SS3 (O'reilly et al., 2014), Sleep-EDF-78 (Kemp et al., 2000) and ISRUC-S1 (Khalighi et al., 2016). According to the AASM standard, we remove the Movement and Unknown stages and merge the S3 and S4 stages into a N3 stage, resulting in five categories of Wake, N1, N2, N3, and REM sleep stages. Each PSG recording is downsampled to 100 Hz and divided into 30-second sleep epochs.

**ISRUC-S3** collects PSG from 10 subjects (one male and nine females) over 10 nights, containing 8589 sleep epochs. We select nine of the 12 channels and remove the leg EMG and ECG channels that are far from the brain. We select three modalities with nine channels (six-channel EEG, two-channel EOG, and one-channel chin EMG). **MASS-SS3** collects PSG from 62 subjects (28 males and 34 females) over 62 nights, containing 59304 sleep epochs. We select three similar modalities with nine channels as ISRUC-S3. **Sleep-EDF-78** collects PSG from 78 subjects (41 males and 37 females) over 153 nights, containing 195479 sleep epochs. We selected three similar modalities with full channels (two-channel EEG, one-channel EOG, and one-channel chin EMG) as ISRUC-S3. **ISRUC-S1** collects PSG from 100 subjects (55 males and 45 females) over 100 nights, containing 87187 sleep epochs. We select the same channels and modalities as ISRUC-S3.

### 5.2 COMPARATIVE EXPERIMENT RESULTS

To demonstrate the advantages of SleepSMC in sleep staging, we compare SleepSMC with a total of 11 methods on both multimodal and unimodal testing scenarios on four datasets.

**Multimodal testing scenario.** We assign uncertainty-based weights to all modalities for contrastive learning and concatenate features from all modalities for joint testing during testing. Table 1 presents the results of the multimodal testing scenario, which indicate that SleepSMC consistently outperforms the other methods across all four datasets. Each method based on multimodal data achieves relatively high performance in sleep staging. However, SleepSMC still improves the $Accuracy$ of the ISRUC-S3 dataset by 1.7% compared to the second-best method, BSTT, with a 2.5% increase in $Macro\ F1$ and a 2.3% increase in $Kappa$. Furthermore, compared to the baseline method, FeatureNet, SleepSMC outperforms by over 3% on all three metrics. The improvements made to the other datasets are also significant.

**Unimodal testing scenario.** We designate one modality as the primary modality and the other two modalities as auxiliary modalities. During training, we train using all modalities collectively,

assigning uncertainty-based weights to the auxiliary modalities. During testing, only the single primary modality is employed for evaluation. Tables 2, (5, 6 and 7 in Appendix A.4) present the results of the unimodal testing scenario, which indicate that SleepSMC consistently outperforms the other methods across all four datasets. Notably, traditional multimodal methods show a sharp accuracy drop with single-modality testing, while SleepSMC remains effective. On the ISRUC-S3 dataset, SleepSMC improves the *Accuracy* in each primary modality experiment by over 1% compared to the second-best method. For the poor-performing primary modality, EMG, it increases by 2.1% in *Accuracy*, with improvements of 1.7% in *Macro F*1 and 2.6% in *Kappa*. On the other datasets, some methods fail to classify N1 and N3 stages, especially when using only EMG from ISRUC-S1, with BSTT suffering from severe prediction imbalance, identifying only one category. In contrast, SleepSMC consistently performs well, with improvements in single-modality performance.

Table 1: The performance comparison of state-of-the-art methods and SleepSMC for **multimodal testing scenario** on four datasets. The **bold** and underline items denote the best and second-best results, respectively.

| Dataset | Method | Overall results | | | F1 for each category | | | | |
|---|---|---|---|---|---|---|---|---|---|
| | | Accuracy | Macro F1 | Kappa | Wake | N1 | N2 | N3 | REM |
| ISRUC-S3 | FeatureNet (Jia et al., 2021b) | 0.7628 | 0.7495 | 0.6975 | 0.8706 | 0.5430 | 0.7152 | 0.8470 | 0.7715 |
| | DeepSleepNet (Supratak et al., 2017) | 0.7426 | 0.7135 | 0.6682 | **0.8788** | 0.4330 | 0.7248 | 0.8262 | 0.7050 |
| | AttnSleep (Eldele et al., 2021) | 0.7656 | 0.7480 | 0.6993 | 0.8620 | 0.5166 | 0.7659 | 0.8754 | 0.7202 |
| | DAN(Tang et al., 2022) | 0.7720 | 0.7431 | 0.7058 | 0.8302 | 0.4441 | **0.7808** | **0.8794** | 0.7808 |
| | BSTT Liu & Jia (2023) | 0.7756 | 0.7568 | 0.7114 | 0.8325 | 0.4901 | 0.7698 | 0.8722 | **0.8193** |
| | XSleepNet Phan et al. (2021) | 0.6705 | 0.6440 | 0.5771 | 0.7888 | 0.4283 | 0.6909 | 0.8045 | 0.5077 |
| | SleepPrintNet (Jia et al., 2020) | 0.7702 | 0.7573 | 0.7043 | 0.8287 | 0.5311 | 0.7725 | 0.8638 | 0.7903 |
| | MMASleepNet (Yubo et al., 2022) | 0.7732 | 0.7343 | 0.7066 | 0.8778 | 0.3950 | 0.7708 | 0.8703 | 0.7576 |
| | SimCLR (Chen et al., 2020) | 0.7738 | 0.7597 | 0.7110 | 0.8771 | 0.5600 | 0.7288 | 0.8488 | 0.7839 |
| | DrFuse Yao et al. (2024) | 0.7741 | 0.7469 | 0.7091 | 0.8748 | 0.4612 | 0.7625 | 0.8658 | 0.7702 |
| | MERL Liu et al. (2024) | 0.7559 | 0.7458 | 0.6876 | 0.8350 | 0.5275 | 0.7432 | 0.8554 | 0.7678 |
| | **Ours** | **0.7930** | **0.7815** | **0.7344** | 0.8765 | **0.5715** | 0.7753 | 0.8724 | 0.8117 |
| MASS-SS3 | FeatureNet (Jia et al., 2021b) | 0.8530 | 0.8042 | 0.7835 | 0.8808 | 0.5318 | 0.8950 | 0.8438 | 0.8695 |
| | DeepSleepNet (Supratak et al., 2017) | 0.8561 | 0.8012 | 0.7862 | 0.8917 | 0.5056 | 0.8997 | 0.8401 | 0.8690 |
| | AttnSleep (Eldele et al., 2021) | 0.8592 | 0.8036 | 0.7914 | 0.8958 | 0.4989 | 0.9007 | 0.8526 | 0.8700 |
| | DAN (Tang et al., 2022) | 0.8034 | 0.6801 | 0.7002 | 0.8050 | 0.1385 | 0.8708 | 0.7999 | 0.7862 |
| | BSTT Liu & Jia (2023) | 0.8114 | 0.7492 | 0.7190 | 0.8292 | 0.4226 | 0.8637 | 0.8023 | 0.8283 |
| | XSleepNet Phan et al. (2021) | 0.8066 | 0.7464 | 0.7158 | 0.8502 | 0.4044 | 0.8660 | 0.7995 | 0.8116 |
| | SleepPrintNet (Jia et al., 2020) | 0.8457 | 0.7855 | 0.7656 | 0.8628 | 0.4895 | 0.8918 | 0.8223 | 0.8612 |
| | MMASleepNet (Yubo et al., 2022) | 0.8627 | 0.7983 | 0.7940 | 0.8873 | 0.4668 | 0.9023 | 0.8457 | 0.8892 |
| | SimCLR (Chen et al., 2020) | 0.8607 | 0.8101 | 0.7931 | 0.8944 | 0.5335 | 0.9008 | 0.8453 | 0.8765 |
| | DrFuse Yao et al. (2024) | 0.8628 | 0.8086 | 0.7964 | 0.8849 | 0.5186 | 0.9028 | **0.8542** | 0.8827 |
| | MERL Liu et al. (2024) | 0.8605 | 0.8055 | 0.7915 | 0.8907 | 0.5200 | 0.8996 | 0.8324 | 0.8851 |
| | **Ours** | **0.8686** | **0.8193** | **0.8058** | **0.9047** | **0.5472** | **0.9049** | 0.8463 | **0.8932** |
| Sleep-EDF-78 | FeatureNet (Jia et al., 2021b) | 0.8012 | 0.7331 | 0.7257 | 0.9195 | 0.3849 | 0.8220 | 0.7807 | 0.7586 |
| | DeepSleepNet (Supratak et al., 2017) | 0.7973 | 0.7392 | 0.7205 | 0.9100 | 0.4271 | 0.8267 | 0.7843 | 0.7477 |
| | AttnSleep (Eldele et al., 2021) | 0.7492 | 0.6912 | 0.6559 | 0.8280 | 0.3815 | 0.8158 | 0.7137 | 0.7170 |
| | DAN (Tang et al., 2022) | 0.7239 | 0.5915 | 0.6057 | 0.8519 | 0.0695 | 0.7629 | 0.7048 | 0.5682 |
| | BSTT Liu & Jia (2023) | 0.7321 | 0.6335 | 0.6245 | 0.8627 | 0.1963 | 0.7587 | 0.7323 | 0.6178 |
| | XSleepNet Phan et al. (2021) | 0.7577 | 0.6855 | 0.6631 | 0.8739 | 0.3589 | 0.7977 | 0.7261 | 0.6709 |
| | SleepPrintNet (Jia et al., 2020) | 0.7849 | 0.7149 | 0.7009 | 0.9060 | 0.3711 | 0.8084 | 0.7541 | 0.7353 |
| | MMASleepNet (Yubo et al., 2022) | 0.7914 | 0.7012 | 0.7078 | 0.9126 | 0.4082 | 0.8186 | 0.6055 | 0.7611 |
| | SimCLR (Chen et al., 2020) | 0.8027 | 0.7352 | 0.7267 | 0.9170 | 0.3937 | 0.8281 | 0.7837 | 0.7532 |
| | DrFuse Yao et al. (2024) | 0.8009 | 0.7411 | 0.7235 | 0.9082 | 0.4189 | 0.8264 | 0.7846 | 0.7674 |
| | MERL Liu et al. (2024) | 0.7990 | 0.7267 | 0.7196 | 0.9135 | 0.3681 | 0.8204 | 0.7841 | 0.7472 |
| | **Ours** | **0.8158** | **0.7558** | **0.7450** | **0.9243** | **0.4285** | **0.8359** | **0.8000** | **0.7905** |
| ISRUC-S1 | BSTT Liu & Jia (2023) | 0.7247 | 0.6890 | 0.6423 | 0.8359 | 0.3622 | 0.7115 | 0.8000 | 0.7356 |
| | XSleepNet Phan et al. (2021) | 0.7444 | 0.7226 | 0.6707 | 0.8688 | 0.4510 | 0.7323 | 0.8083 | 0.7526 |
| | DrFuse Yao et al. (2024) | 0.7441 | 0.7215 | 0.6669 | 0.8143 | 0.4791 | 0.7310 | 0.7978 | 0.7852 |
| | MERL Liu et al. (2024) | 0.7245 | 0.7042 | 0.6417 | 0.7954 | 0.4714 | 0.7086 | 0.7869 | 0.7588 |
| | **Ours** | **0.7710** | **0.7462** | **0.7018** | **0.8862** | **0.4795** | **0.7510** | **0.8138** | **0.8004** |

## 5.3 ABLATION EXPERIMENT RESULTS

To further illustrate the contributions of each module in SleepSMC, we performed ablation experiments on the Uncertainty-based Feature Weighting and Supervised Modality-level Instance Contrastive Coordination modules. As shown in Table 3, the results of multimodal and unimodal testing scenarios on three datasets demonstrate the effectiveness of each module. The Supervised Modality-level Contrastive Coordination module plays a more significant role in both scenarios. Meanwhile, the Uncertainty-based Feature Weighting module demonstrates relatively enhanced performance in

Table 2: The performance comparison of state-of-the-art methods and SleepSMC on the **ISRUC-S3** dataset for **unimodal testing scenario**. The **bold** and underline items denote the best and second-best results, respectively.

| Modality | Method | Overall results | | | F1 for each category | | | | |
|---|---|---|---|---|---|---|---|---|---|
| | | Accuracy | Macro F1 | Kappa | Wake | N1 | N2 | N3 | REM |
| EEG | FeatureNet (Jia et al., 2021b) | 0.7277 | 0.7104 | 0.6510 | 0.8806 | 0.5064 | 0.6726 | 0.8006 | 0.6917 |
| | DeepSleepNet (Supratak et al., 2017) | 0.6904 | 0.6507 | 0.6057 | 0.8641 | 0.3296 | 0.6349 | 0.7959 | 0.6289 |
| | AttnSleep (Eldele et al., 2021) | 0.7338 | 0.7105 | 0.6592 | 0.8581 | 0.4636 | 0.7320 | 0.8524 | 0.6463 |
| | DAN(Tang et al., 2022) | 0.7212 | 0.6791 | 0.6400 | 0.8077 | 0.3511 | 0.7352 | **0.8686** | 0.6328 |
| | BSTT Liu & Jia (2023) | 0.7191 | 0.6921 | 0.6371 | 0.8061 | 0.4312 | 0.6989 | 0.8502 | 0.6742 |
| | XSleepNet Phan et al. (2021) | 0.6555 | 0.6322 | 0.5614 | 0.8525 | 0.4562 | 0.6225 | 0.8015 | 0.4281 |
| | SleepPrintNet (Jia et al., 2020) | 0.5459 | 0.4862 | 0.3924 | 0.5109 | 0.3404 | 0.6161 | 0.6669 | 0.2968 |
| | MMASleepNet (Yubo et al., 2022) | 0.6313 | 0.5975 | 0.5150 | 0.7815 | 0.3486 | 0.6771 | 0.6471 | 0.5333 |
| | SimCLR (Chen et al., 2020) | 0.7338 | 0.7163 | 0.6598 | 0.8777 | 0.4978 | 0.6883 | 0.8260 | 0.6915 |
| | DrFuse Yao et al. (2024) | 0.7532 | 0.7138 | 0.6818 | 0.8780 | 0.3872 | **0.7794** | 0.8609 | 0.6636 |
| | MERL Liu et al. (2024) | 0.7467 | 0.7295 | 0.6758 | 0.8524 | **0.5212** | 0.7328 | 0.8603 | 0.6808 |
| | **Ours** | **0.7646** | **0.7397** | **0.6969** | **0.8882** | 0.5069 | 0.7467 | 0.8636 | **0.6932** |
| EOG | FeatureNet (Jia et al., 2021b) | 0.7210 | 0.6932 | 0.6399 | 0.7995 | 0.4640 | 0.7196 | 0.8570 | 0.6259 |
| | DeepSleepNet (Supratak et al., 2017) | 0.7234 | 0.6902 | 0.6388 | 0.8142 | 0.4145 | 0.7100 | 0.8541 | 0.6583 |
| | AttnSleep (Eldele et al., 2021) | 0.7226 | 0.6992 | 0.6416 | 0.8248 | 0.4608 | 0.7115 | 0.8591 | 0.6399 |
| | DAN(Tang et al., 2022) | 0.7136 | 0.6647 | 0.6288 | 0.7733 | 0.2902 | **0.7406** | 0.8652 | 0.6542 |
| | BSTT Liu & Jia (2023) | 0.4700 | 0.3163 | 0.2790 | 0.1169 | 0.2352 | 0.5895 | 0.6400 | 0.0000 |
| | XSleepNet Phan et al. (2021) | 0.6288 | 0.6071 | 0.5233 | 0.6958 | 0.3684 | 0.6572 | 0.7882 | 0.5260 |
| | SleepPrintNet (Jia et al., 2020) | 0.3745 | 0.2531 | 0.1788 | 0.3553 | 0.0239 | 0.5680 | 0.0000 | 0.3183 |
| | MMASleepNet (Yubo et al., 2022) | 0.2096 | 0.1745 | 0.0619 | 0.2750 | 0.2712 | 0.0000 | 0.0000 | 0.3264 |
| | SimCLR (Chen et al., 2020) | 0.7246 | 0.7007 | 0.6458 | 0.8097 | **0.4788** | 0.7096 | 0.8523 | 0.6529 |
| | DrFuse Yao et al. (2024) | 0.6947 | 0.6799 | 0.6078 | 0.7522 | 0.4579 | 0.7115 | 0.8317 | 0.6460 |
| | MERL Liu et al. (2024) | 0.6976 | 0.6741 | 0.6132 | 0.7996 | 0.3912 | 0.6808 | 0.8351 | **0.6640** |
| | **Ours** | **0.7444** | **0.7168** | **0.6697** | **0.8386** | 0.4765 | 0.7360 | **0.8722** | 0.6607 |
| EMG | FeatureNet (Jia et al., 2021b) | 0.4040 | 0.3731 | 0.2214 | 0.5326 | 0.1181 | 0.4282 | 0.3794 | 0.4075 |
| | DeepSleepNet (Supratak et al., 2017) | 0.4166 | 0.3704 | 0.2404 | 0.5597 | 0.0403 | 0.4548 | 0.3978 | 0.3992 |
| | AttnSleep (Eldele et al., 2021) | 0.3915 | 0.3814 | 0.2191 | 0.5096 | 0.2067 | 0.3804 | 0.4152 | 0.3950 |
| | DAN(Tang et al., 2022) | 0.4048 | 0.3381 | 0.2267 | 0.5541 | 0.0065 | **0.4670** | 0.2262 | 0.4365 |
| | BSTT Liu & Jia (2023) | 0.3046 | 0.0934 | 0.0000 | 0.0000 | 0.0000 | 0.4669 | 0.0000 | 0.0000 |
| | XSleepNet Phan et al. (2021) | 0.3660 | 0.3484 | 0.1935 | 0.4519 | 0.1654 | 0.3665 | 0.3833 | 0.3748 |
| | SleepPrintNet (Jia et al., 2020) | 0.3319 | 0.2313 | 0.0939 | 0.4214 | 0.0359 | 0.4327 | 0.0000 | 0.2667 |
| | MMASleepNet (Yubo et al., 2022) | 0.2517 | 0.1969 | 0.1062 | 0.4155 | 0.1450 | 0.0000 | 0.0000 | 0.4240 |
| | SimCLR (Chen et al., 2020) | 0.4177 | 0.3906 | 0.2435 | 0.5605 | 0.1397 | 0.4303 | 0.4258 | 0.3968 |
| | DrFuse Yao et al. (2024) | 0.3857 | 0.3789 | 0.2318 | **0.6026** | 0.1923 | 0.3549 | 0.3272 | 0.4176 |
| | MERL Liu et al. (2024) | 0.3981 | 0.3907 | 0.2348 | 0.4875 | **0.2077** | 0.3879 | 0.4008 | **0.4696** |
| | **Ours** | **0.4384** | **0.4075** | **0.2693** | 0.5868 | 0.1281 | 0.4404 | **0.4301** | 0.4523 |

unimodal than multimodal scenarios. This suggests that when only the primary modality is available during testing, the quality of auxiliary modalities during training becomes more critical, making feature weighting essential.

In addition, in Table 8, we conduct experiments on the uncertainty variance metric, where we compare the direct application of the negative value of Eq.5, $-r_i^{u_a}$, with the effect of using Eq.6, which incorporates an exponential function for smoothing and scaling, thereby demonstrating the necessity of the latter.

Table 3: Ablation study of Uncertainty-based Feature Weighting and Supervised Modality-level instance Contrastive Coordination modules of SleepSMC(✓✓) on three public datasets. The **bold** and underline items denote the best and second-best results, respectively.

| Dataset | U | C | Accuracy / Macro F1 / Kappa | | | |
|---|---|---|---|---|---|---|
| | | | multimodal | EEG | EOG | EMG |
| ISRUC-S3 | × | × | 0.7628 / 0.7495 / 0.6975 | 0.7277 / 0.7104 / 0.6510 | 0.7210 / 0.6932 / 0.6399 | 0.4040 / 0.3731 / 0.2214 |
| | × | ✓ | 0.7918 / 0.7767 / 0.7329 | 0.7609 / 0.7358 / 0.6932 | 0.7343 / 0.7087 / 0.6570 | 0.4209 / 0.3974 / 0.2490 |
| | ✓ | ✓ | **0.7930 / 0.7815 / 0.7344** | **0.7646 / 0.7397 / 0.6969** | **0.7444 / 0.7168 / 0.6697** | **0.4384 / 0.4075 / 0.2693** |
| MASS-SS3 | × | × | 0.8530 / 0.8042 / 0.7835 | 0.8366 / 0.7571 / 0.7575 | 0.7970 / 0.7102 / 0.6939 | 0.5327 / 0.3664 / 0.2439 |
| | × | ✓ | 0.8680 / 0.8185 / 0.8045 | 0.8455 / 0.7725 / 0.7703 | 0.8148 / 0.7449 / 0.7249 | 0.5339 / 0.3679 / 0.2463 |
| | ✓ | ✓ | **0.8686 / 0.8193 / 0.8058** | **0.8517 / 0.7871 / 0.7798** | **0.8227 / 0.7534 / 0.7359** | **0.5408 / 0.3770 / 0.2613** |
| Sleep-EDF-78 | × | × | 0.8012 / 0.7331 / 0.7257 | 0.7772 / 0.6944 / 0.6928 | 0.7238 / 0.6375 / 0.6159 | 0.5180 / 0.3076 / 0.2849 |
| | × | ✓ | 0.8146 / 0.7552 / 0.7434 | 0.7894 / 0.7162 / 0.7088 | 0.7250 / 0.6394 / 0.6172 | 0.5211 / 0.3159 / 0.2908 |
| | ✓ | ✓ | **0.8158 / 0.7558 / 0.7450** | **0.7959 / 0.7217 / 0.7169** | **0.7311 / 0.6494 / 0.6258** | **0.5293 / 0.3167 / 0.3017** |

## 5.4 INTERPRETABLE ANALYSIS RESULTS

**Weight Visualization Analysis.** As shown in Figure 3, we visualize the uncertainty-based weight variations of auxiliary modalities under different primary modalities during training. From Figure 3 (a) and (b), it can be observed that despite the lower accuracy of EMG-based sleep staging, SleepSMC effectively adjusts the weight of EMG during contrastive learning due to its robustness (lower uncertainty), resulting in significant improvements. This demonstrates that prioritizing robustness over accuracy in supervised contrastive learning is crucial for performance enhancement.

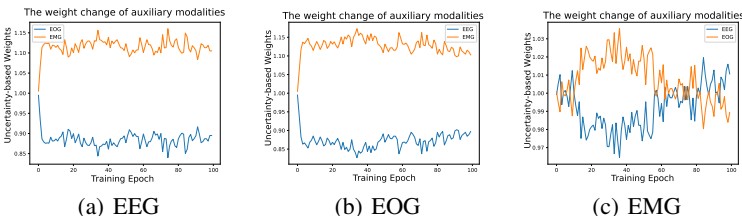

(a) EEG        (b) EOG        (c) EMG

Figure 3: Visualization of the uncertainty-based weight changes for the two auxiliary modalities with the primary modalities: (a) EEG, (b) EOG, and (c) EMG.

**Feature Visualization Analysis.** We exploit t-SNE (Van der Maaten & Hinton, 2008) to visualize the feature embeddings of each primary modality for both FeatureNet and SleepSMC methods. As shown in Figure 4, regardless of which primary modality is employed for testing, SleepSMC consistently displays more explicit and compact classification boundaries. Notably, for the most challenging N1 stage, the distribution improves from being completely scattered to having a clear boundary. Although EMG-based sleep staging generally performs poorly, SleepSMC still significantly improves over FeatureNet and achieves a clear separation for categories like Wake and REM.

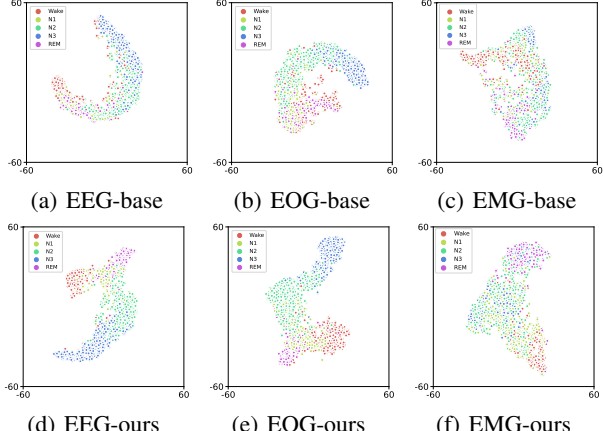

(a) EEG-base      (b) EOG-base      (c) EMG-base

(d) EEG-ours      (e) EOG-ours      (f) EMG-ours

Figure 4: Visualization of the t-SNE embeddings for each modality of the ISRUC-S3 dataset using (a)(b)(c) FeatureNet and (d)(e)(f) SleepSMC.

## 6 CONCLUSION

We proposed SleepSMC, a framework that combines supervised modality-level instance contrastive coordination with uncertainty-based feature weighting for ubiquitous sleep staging. SleepSMC enhances the performance of the primary modality by leveraging supervised modality-level instance contrastive coordination during training and dynamically adjusting the contributions of auxiliary modality features based on uncertainty. Experiments on four public datasets show that SleepSMC achieves superior cross-subject performance, even when only a single modality is employed during testing. The framework effectively captures category-related relationships across and within modalities, demonstrating its robustness, interpretability, and potential for real-world applications.

## ACKNOWLEDGEMENT

This work is supported by the Natural Science Foundation of China (No.62302487), Improvement Project of Chinese Academy of Sciences (No.GSZXKYZB2025007), the Science and Technology Innovation Program of Hunan Province (No.2022RC4006, No.2024JJ9031), and AI Dream Project (No.SZSM202402).

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

# A    APPENDIX

## A.1    SYMBOL DEFINITION

Table 4: List of mathematical notations used in the SleepSMC method.

| Notation | Definition |
|---|---|
| $D_{train}, D_{test}$ | Training set and cross-subject testing set. |
| $x_i, x_j$ | Input PSG signal of the $i$-th and $j$-th sample, respectively. |
| $y_i, y_j$ | Ground truth sleep stage label for the $i$-th and $j$-th sample. |
| $U$ | Total number of modalities. |
| $I, J$ | Number of samples in the training and testing sets, respectively. |
| $K$ | Total number of sleep stage categories. |
| $u \in \{e, o, m\}$ | Modality index: $e$ for EEG, $o$ for EOG, $m$ for EMG. |
| $x_i^u$ | Modality-specific input signal of the $i$-th sample for modality $u$. |
| $f_i^u$ | Features extracted for modality $u$ from the $i$-th sample. |
| $G_f^u, G_y^u$ | Feature extractor and classifier for modality $u$. |
| $\theta_f^u$ | Parameters of the feature extractor for modality $u$. |
| $u_p$ | Primary modality (used during testing). |
| $u_a$ | Auxiliary modality (used only during training). |
| $T$ | Number of random masks for uncertainty estimation. |
| $\phi(x, \eta)$ | Random mask transformation applied to input $x$ with ratio $\eta$. |
| $p_i^u$ | Confidence score for the predicted category from modality $u$. |
| $r_i^u$ | Uncertainty variance of the $i$-th sample for modality $u$. |
| $w_i^u$ | Weight assigned to the features of modality $u$ for the $i$-th sample. |
| $\tilde{f}_{i'}^u$ | L2-normalized feature of modality $u$ for the $i$-th sample. |
| $L_{cls}^u$ | Modality-specific classification loss for modality $u$. |
| $L_{Con}$ | Supervised contrastive loss. |
| $L_{total}$ | Total loss combining classification and contrastive losses. |

## A.2    METHOD IMPLEMENT

The pipeline of SleepSMC is shown in Algorithm 1. During training, the model performs modality-specific feature extraction for each instance and applies uncertainty-based feature weighting for auxiliary modalities. The model parameters are then updated by optimizing the modality-specific classification loss and supervised contrastive loss. During testing, only the primary modality (e.g., EEG) is employed to extract features from cross-subject test data, and the trained classifier is employed to predict the sleep stages. The final output is the predicted results for the testing set.

---

**Algorithm 1** SleepSMC: End-to-End Training and Testing

---

**Input:** Training data $\mathcal{D}_{train} = \{(x_i, y_i)\}_{i=1}^{I}$, testing data $\mathcal{D}_{test} = \{x_j^e\}_{j=1}^{J}$
**Output:** Predicted results $\hat{y}_j$ for $\mathcal{D}_{test}$
 1: **Initialization:** Initialize model parameters $\theta^f$ for feature extractors and $\theta^y$ for classifiers $G_y$
 2: **Training:**
 3: **For** not converged **do**
 4:    **for** $i = 1$ to $I$ **do**                                  ▷ Modality-Specific Feature Extraction
 5:       Extract modality-specific features:
 6:       $f_i^e = G_f^e(x_i^e); \quad f_i^o = G_f^o(x_i^o); \quad f_i^m = G_f^m(x_i^m)$
 7:       **for** each auxiliary modality $u_a \in \{e, o, m\} \setminus u_p$ **do**    ▷ Uncertainty-Based Feature Weighting
 8:          Estimate uncertainty $r_i^{u_a}$ using Eq. 5
 9:          Compute feature weight $w_i^{u_a}$ using Eq. 7
10:          Weight auxiliary features: $f_i^{u_a'} = w_i^{u_a} \cdot f_i^{u_a}$
11:       **end for**
12:       Normalize features $\widetilde{f}_i^u = \left\{ \frac{f_i^{u_a'}}{\|f_i^{u_a'}\|}, \frac{f_i^{u_p}}{\|f_i^{u_p}\|} \right\}$
13:    **end for**
14:    Compute modality-specific classification loss $\mathcal{L}_{\text{cls}}^u$ using Eq. 1
15:    Compute supervised contrastive loss $\mathcal{L}_{\text{Con}}$ using Eq. 9      ▷ Supervised Contrastive Learning
16:    Compute total loss: $\mathcal{L}_{\text{total}} = \sum_u \mathcal{L}_{\text{cls}}^u + \mathcal{L}_{\text{Con}}$
17:    Update model parameters $\theta^f, \theta^y$ using backpropagation
18: **end For**
19:
20: **Testing:**
21: **for** $j = 1$ to $J$ **do**                              ▷ Predicting on test data after training
22:    Extract features from cross-subject test data: $f_j^e = G_f^e(x_j^e)$        ▷ Take EEG as an example
23:    Predict sleep stage $\hat{y}_j = \arg\max_k \left[ G_y^e(f_j^e) \right]_k$
24: **end for**
25: **Return:** Predicted results $\hat{y}_j$ for $\mathcal{D}_{test}$

---

## A.3 EXPERIMENT SETTINGS AND IMPLEMENTATION

### A.3.1 PARAMATER SETTINGS AND METRICS

In the experiment, we use FeatureNet (Jia et al., 2021b) as the backbone of SleepSMC to extract modality-specific features. The subjects are divided into five domains for five-fold cross-validation, with each domain serving as the testing set and the remaining four as the training set. From each training set, 20% is set aside as a validation set for model selection. The best model is saved and evaluated on the corresponding cross-subject testing set, and the final results are averaged across all five folds for an overall evaluation. The model is trained for 100 epochs using the Adam optimizer with a learning rate 0.001. The temperature $\tau$ for contrastive learning is set to 0.1. The number of random masks $T$ is set to 9, with a random masking ratio $\eta$ of 15% on data.

The evaluation metrics include $Accuracy$, $Macro\ F1$, $Kappa$, and the $F1$ score for each category. $Accuracy$ represents the proportion of correctly classified instances. The $F1$ score combines precision and recall for each category, while $Macro\ F1$ averages the $F1$ scores across all sleep stages. $Kappa$ measures the agreement between predictions and ground truth, considering random chance.

### A.3.2 BASELINE METHODS AND SETTINGS

We select 11 classical and state-of-the-art sleep staging methods as baselines: six conventional (FeatureNet, DeepSleepNet, AttnSleep, DAN, BSTT, XSleepNet), two multimodal (SleepPrintNet, MMASleepNet) and three multimodal coordination methods(SimCLR, DrFuse, MERL). In multimodal and multimodal coordination methods, multiple modalities are employed as inputs during training, but only a single modality is employed during testing, with missing modalities filled with zeros. Among these, FeatureNet (Jia et al., 2021b) utilizes a dual-scale CNN architecture with varying kernel sizes to perform modality-specific feature extraction, followed by fully connected layers to classify fused features. DeepSleepNet (Supratak et al., 2017) combines CNNs for local pattern and spatial relationship extraction with BiLSTM for modeling long-term dependencies. AttnSleep (Eldele et al., 2021) integrates a multi-resolution CNN, adaptive feature recalibration for learning feature interdependencies, and multi-head attention for temporal modeling. DAN (Tang

et al., 2022) is a CNN-BiGRU model that employs MMD alignment to learn domain-invariant features. BSTT (Liu & Jia, 2023) uses Bayesian inference to learn spatial and temporal relationships, combining Transformer and fully connected networks, with KL divergence enforcing consistency between prior and posterior distributions. XSleepNet Phan et al., 2021 combines CNN, GRU, and attention mechanisms to learn spatial and temporal features from EEG signals, using multi-branch architectures for joint classification. SleepPrintNet (Jia et al., 2020) is a multimodal method using 1D CNN for modality-specific feature extraction, focusing on temporal features across all modalities and spectral-spatial features for EEG. MMASleepNet (Yubo et al., 2022) combines 1D CNN and squeeze-and-excitation for modality-specific learning, with transformer encoders for feature fusion. SimCLR (Chen et al., 2020) uses FeatureNet for modality-specific learning and classification, treating pairs of modalities from the same synchronized samples as positive pairs and others as negative pairs. DrFuse (Yao et al., 2024) combines shared and modality-specific models to learn joint representations from multimodal inputs. MERL Liu et al. (2024) achieves intra-modality and inter-modality alignment through contrastive learning, effectively learning and sharing modality representations.

## A.4 SUPPLEMENTARY EXPERIMENT RESULTS

Table 5: The performance comparison of state-of-the-art methods and SleepSMC on **MASS-SS3** dataset for the **unimodal testing scenario**. The **bold** and underline items denote the best and second-best results, respectively.

| Modality | Method | Overall results | | | F1 for each category | | | | |
|---|---|---|---|---|---|---|---|---|---|
| | | Accuracy | Macro F1 | Kappa | Wake | N1 | N2 | N3 | REM |
| EEG | FeatureNet (Jia et al., 2021b) | 0.8366 | 0.7571 | 0.7575 | 0.8813 | 0.3558 | 0.8943 | 0.8302 | 0.8238 |
| | DeepSleepNet (Supratak et al., 2017) | 0.8348 | 0.7456 | 0.7513 | 0.8811 | 0.3028 | 0.8911 | 0.8367 | 0.8160 |
| | AttnSleep (Eldele et al., 2021) | 0.8320 | 0.7512 | 0.7489 | 0.8801 | 0.3483 | 0.8918 | 0.8231 | 0.8129 |
| | DAN (Tang et al., 2022) | 0.7976 | 0.6556 | 0.6914 | 0.8301 | 0.0318 | 0.8734 | 0.8029 | 0.7400 |
| | BSTT Liu & Jia (2023) | 0.7927 | 0.6855 | 0.6856 | 0.8183 | 0.1978 | 0.8650 | 0.8141 | 0.7321 |
| | XSleepNet Phan et al. (2021) | 0.7973 | 0.7175 | 0.7000 | 0.8514 | 0.3134 | 0.8619 | 0.7935 | 0.7674 |
| | SleepPrintNet (Jia et al., 2020) | 0.7203 | 0.5159 | 0.5591 | 0.3086 | 0.0000 | 0.8518 | 0.8267 | 0.5925 |
| | MMASleepNet (Yubo et al., 2022) | 0.7997 | 0.6587 | 0.6885 | 0.7988 | 0.0449 | 0.8706 | 0.8376 | 0.7415 |
| | SimCLR (Chen et al., 2020) | 0.8350 | 0.7586 | 0.7577 | 0.8913 | 0.3653 | 0.8893 | 0.8267 | 0.8204 |
| | DrFuse Yao et al. (2024) | 0.8375 | 0.7513 | 0.7573 | 0.8684 | 0.3278 | 0.8995 | 0.8455 | 0.8153 |
| | MERL Liu et al. (2024) | 0.8330 | 0.7659 | 0.7508 | 0.8768 | 0.4255 | 0.8888 | 0.8110 | 0.8276 |
| | **Ours** | **0.8517** | **0.7871** | **0.7798** | **0.8955** | **0.4478** | **0.9046** | **0.8502** | **0.8376** |
| EOG | FeatureNet (Jia et al., 2021b) | 0.7970 | 0.7102 | 0.6939 | 0.8056 | 0.3223 | 0.8647 | 0.8022 | 0.7562 |
| | DeepSleepNet (Supratak et al., 2017) | 0.8025 | 0.7125 | 0.7036 | 0.7940 | 0.3110 | 0.8689 | 0.8053 | 0.7833 |
| | AttnSleep (Eldele et al., 2021) | 0.8006 | 0.7044 | 0.6966 | 0.8037 | 0.2878 | 0.8673 | 0.7919 | 0.7715 |
| | DAN (Tang et al., 2022) | 0.7478 | 0.5804 | 0.6137 | 0.6088 | 0.0000 | 0.8600 | 0.7672 | 0.6662 |
| | BSTT Liu & Jia (2023) | 0.7740 | 0.6956 | 0.6608 | 0.7291 | 0.3846 | 0.8627 | 0.7733 | 0.7281 |
| | XSleepNet Phan et al. (2021) | 0.7804 | 0.6984 | 0.6721 | 0.7754 | 0.3494 | 0.8516 | 0.7587 | 0.7570 |
| | SleepPrintNet (Jia et al., 2020) | 0.1891 | 0.0922 | 0.0239 | 0.1285 | 0.0008 | 0.0000 | 0.0000 | 0.3317 |
| | MMASleepNet (Yubo et al., 2022) | 0.2988 | 0.1826 | 0.1235 | 0.1887 | 0.0000 | 0.3139 | 0.0018 | 0.4087 |
| | SimCLR (Chen et al., 2020) | 0.8000 | 0.7098 | 0.6993 | 0.7840 | 0.3262 | 0.8754 | 0.8084 | 0.7548 |
| | DrFuse Yao et al. (2024) | 0.7805 | 0.7072 | 0.6712 | 0.7659 | 0.3973 | 0.8653 | 0.7635 | 0.7437 |
| | MERL Liu et al. (2024) | 0.7941 | 0.7142 | 0.6948 | 0.8025 | 0.3433 | 0.8622 | 0.7792 | 0.7835 |
| | **Ours** | **0.8227** | **0.7534** | **0.7359** | **0.8358** | **0.4329** | **0.8854** | **0.8099** | **0.8031** |
| EMG | FeatureNet (Jia et al., 2021b) | 0.5327 | 0.3664 | 0.2439 | 0.4860 | 0.1469 | 0.6539 | 0.0680 | 0.4773 |
| | DeepSleepNet (Supratak et al., 2017) | 0.5232 | 0.2176 | 0.0815 | 0.2376 | 0.0097 | 0.6790 | 0.0000 | 0.1619 |
| | AttnSleep (Eldele et al., 2021) | 0.5267 | 0.2708 | 0.1372 | 0.3196 | 0.0435 | 0.6751 | 0.0113 | 0.3048 |
| | DAN (Tang et al., 2022) | 0.5126 | 0.2651 | 0.1294 | 0.4137 | 0.0233 | 0.6606 | 0.0000 | 0.2276 |
| | BSTT Liu & Jia (2023) | 0.5187 | 0.2137 | 0.1157 | 0.0040 | 0.0000 | **0.6842** | 0.0652 | 0.3150 |
| | XSleepNet Phan et al. (2021) | 0.5252 | 0.3062 | 0.1812 | 0.4361 | 0.0642 | 0.6628 | 0.0003 | 0.3676 |
| | SleepPrintNet (Jia et al., 2020) | 0.2280 | 0.1840 | 0.1133 | 0.2552 | 0.1762 | 0.0000 | 0.0000 | 0.4887 |
| | MMASleepNet (Yubo et al., 2022) | 0.3062 | 0.2293 | 0.1296 | 0.3235 | 0.0928 | 0.2765 | 0.0000 | 0.4536 |
| | SimCLR (Chen et al., 2020) | 0.5287 | 0.3664 | 0.2431 | 0.4929 | 0.1389 | 0.6484 | 0.0783 | 0.4734 |
| | DrFuse Yao et al. (2024) | 0.5059 | 0.3597 | 0.2216 | **0.4999** | 0.1808 | 0.6374 | 0.1443 | 0.4760 |
| | MERL Liu et al. (2024) | 0.5084 | 0.3761 | 0.2346 | 0.3470 | **0.2237** | 0.6296 | **0.1545** | **0.5257** |
| | **Ours** | **0.5408** | **0.3770** | **0.2613** | 0.4991 | 0.1220 | 0.6578 | 0.0988 | 0.5069 |

Table 6: The performance comparison of state-of-the-art methods and SleepSMC on **Sleep-EDF-78** dataset for the **unimodal testing scenario**. The **bold** and underline items denote the best and second-best results, respectively.

| Modality | Method | Overall results | | | F1 for each category | | | | |
|---|---|---|---|---|---|---|---|---|---|
| | | Accuracy | Macro F1 | Kappa | Wake | N1 | N2 | N3 | REM |
| EEG | FeatureNet (Jia et al., 2021b) | 0.7772 | 0.6944 | 0.6928 | 0.9123 | 0.3039 | 0.8154 | 0.7860 | 0.6542 |
| | DeepSleepNet (Supratak et al., 2017) | 0.7588 | 0.6740 | 0.6655 | 0.8829 | 0.2554 | 0.8187 | **0.8011** | 0.6118 |
| | AttnSleep (Eldele et al., 2021) | 0.7789 | 0.7069 | 0.6947 | 0.8996 | 0.3425 | **0.8289** | 0.7830 | 0.6803 |
| | DAN (Tang et al., 2022) | 0.7111 | 0.5638 | 0.5876 | 0.8788 | 0.0584 | 0.7685 | 0.7029 | 0.4104 |
| | BSTT Liu & Jia (2023) | 0.7320 | 0.6503 | 0.6266 | 0.8555 | 0.3101 | 0.7632 | 0.6426 | 0.6801 |
| | XSleepNet Phan et al. (2021) | 0.7685 | 0.6812 | 0.6782 | 0.9038 | 0.3049 | 0.8141 | 0.7420 | 0.6409 |
| | SleepPrintNet (Jia et al., 2020) | 0.6343 | 0.5500 | 0.4975 | 0.7610 | 0.1300 | 0.7233 | 0.7093 | 0.4266 |
| | MMASleepNet (Yubo et al., 2022) | 0.6811 | 0.5124 | 0.5416 | 0.8356 | 0.0531 | 0.7488 | 0.4435 | 0.4810 |
| | SimCLR (Chen et al., 2020) | 0.7824 | 0.6941 | 0.6974 | 0.9170 | 0.2931 | 0.8187 | 0.7919 | 0.6496 |
| | DrFuse Yao et al. (2024) | 0.7753 | 0.6880 | 0.6853 | 0.9018 | 0.2947 | 0.8200 | 0.7948 | 0.6289 |
| | MERL Liu et al. (2024) | 0.7660 | 0.6848 | 0.6770 | 0.9082 | 0.3168 | 0.7956 | 0.7495 | 0.6538 |
| | **Ours** | **0.7959** | **0.7217** | **0.7169** | **0.9197** | **0.3581** | 0.8280 | 0.7991 | **0.7034** |
| EOG | FeatureNet (Jia et al., 2021b) | 0.7238 | 0.6375 | 0.6159 | 0.8468 | 0.2920 | 0.7587 | 0.6259 | 0.6641 |
| | DeepSleepNet (Supratak et al., 2017) | 0.7192 | 0.6240 | 0.6041 | 0.8474 | 0.2472 | 0.7509 | **0.6667** | 0.6075 |
| | AttnSleep (Eldele et al., 2021) | 0.7231 | 0.6395 | 0.6139 | **0.8561** | 0.2939 | 0.7529 | 0.6475 | 0.6470 |
| | DAN (Tang et al., 2022) | 0.6538 | 0.4130 | 0.4890 | 0.8298 | 0.0010 | 0.7020 | 0.0836 | 0.4484 |
| | BSTT Liu & Jia (2023) | 0.7110 | 0.6314 | 0.5995 | 0.8348 | 0.3112 | 0.7482 | 0.6203 | 0.6422 |
| | XSleepNet Phan et al. (2021) | 0.7053 | 0.6149 | 0.5888 | 0.8215 | 0.2576 | 0.7564 | 0.6327 | 0.6061 |
| | SleepPrintNet (Jia et al., 2020) | 0.5710 | 0.4182 | 0.3979 | 0.6571 | 0.2702 | 0.6854 | 0.0000 | 0.4784 |
| | MMASleepNet (Yubo et al., 2022) | 0.6076 | 0.4518 | 0.4463 | 0.7038 | 0.2826 | 0.6815 | 0.0332 | 0.5577 |
| | SimCLR (Chen et al., 2020) | 0.7239 | 0.6334 | 0.6156 | 0.8467 | 0.2713 | 0.7582 | 0.6250 | 0.6656 |
| | DrFuse Yao et al. (2024) | 0.6807 | 0.5692 | 0.5570 | 0.8328 | **0.3548** | 0.7280 | 0.2902 | 0.6404 |
| | MERL Liu et al. (2024) | 0.6855 | 0.6034 | 0.5678 | 0.8179 | 0.2831 | 0.7323 | 0.5706 | 0.6130 |
| | **Ours** | **0.7311** | **0.6494** | **0.6258** | 0.8561 | 0.3140 | **0.7618** | 0.6409 | **0.6744** |
| EMG | FeatureNet (Jia et al., 2021b) | 0.5180 | 0.3076 | 0.2849 | 0.6710 | 0.0241 | 0.5407 | 0.0198 | 0.2825 |
| | DeepSleepNet (Supratak et al., 2017) | 0.5157 | 0.2873 | 0.2735 | 0.6714 | 0.0175 | 0.5419 | 0.0000 | 0.2056 |
| | AttnSleep (Eldele et al., 2021) | 0.5180 | 0.2833 | 0.2749 | 0.6760 | 0.0205 | 0.5528 | 0.0156 | 0.1514 |
| | DAN (Tang et al., 2022) | 0.4913 | 0.2615 | 0.2333 | 0.6429 | 0.0000 | 0.5035 | 0.0000 | 0.1613 |
| | BSTT Liu & Jia (2023) | 0.5147 | 0.3026 | 0.2786 | 0.6661 | 0.0210 | 0.5389 | 0.0173 | 0.2696 |
| | XSleepNet Phan et al. (2021) | 0.5242 | 0.3095 | 0.2930 | 0.6781 | 0.0234 | 0.5392 | 0.0011 | 0.3059 |
| | SleepPrintNet (Jia et al., 2020) | 0.3563 | 0.2096 | 0.0497 | 0.4970 | 0.0237 | 0.2921 | 0.0000 | 0.2350 |
| | MMASleepNet (Yubo et al., 2022) | 0.3779 | 0.2254 | 0.0736 | 0.4740 | 0.0242 | 0.4067 | 0.0000 | 0.2219 |
| | SimCLR (Chen et al., 2020) | 0.5182 | 0.2935 | 0.2805 | 0.6753 | 0.0219 | 0.5412 | 0.0000 | 0.2288 |
| | DrFuse Yao et al. (2024) | 0.4058 | 0.2899 | 0.1573 | 0.5439 | 0.1177 | 0.4309 | **0.0687** | 0.2881 |
| | MERL Liu et al. (2024) | 0.4208 | 0.3080 | 0.2025 | 0.5734 | **0.1459** | 0.3884 | 0.0042 | **0.4281** |
| | **Ours** | **0.5293** | **0.3167** | **0.3017** | **0.6814** | 0.0219 | **0.5529** | 0.0204 | 0.3071 |

Table 7: The performance comparison of state-of-the-art methods and SleepSMC on the **ISRUC-S1** dataset for **unimodal testing scenario**. The **bold** and underline items denote the best and second-best results, respectively.

| ISRUC-S1 | Method | Overall results | | | F1 for each category | | | | |
|---|---|---|---|---|---|---|---|---|---|
| | | Accuracy | Macro F1 | Kappa | Wake | N1 | N2 | N3 | REM |
| EEG | BSTT Liu & Jia (2023) | 0.6840 | 0.6367 | 0.5878 | 0.8060 | 0.2983 | 0.6794 | 0.7881 | 0.6116 |
| | XSleepNet Phan et al. (2021) | 0.7092 | 0.6735 | 0.6233 | 0.8407 | **0.4051** | 0.7122 | 0.8078 | 0.6019 |
| | DrFuse Yao et al. (2024) | 0.6978 | 0.6620 | 0.6087 | 0.7317 | 0.3883 | **0.7208** | 0.7893 | 0.6797 |
| | MERL Liu et al. (2024) | 0.7096 | 0.6690 | 0.6223 | 0.8034 | 0.3618 | 0.7042 | 0.7917 | 0.6837 |
| | **Ours** | **0.7328** | **0.6959** | **0.6536** | **0.8664** | 0.4035 | 0.7145 | **0.8085** | **0.6865** |
| EOG | BSTT Liu & Jia (2023) | 0.3123 | 0.1115 | 0.0019 | 0.0807 | 0.0000 | 0.4767 | 0.0001 | 0.0000 |
| | XSleepNet Phan et al. (2021) | 0.6320 | 0.6047 | 0.5229 | 0.7025 | 0.3460 | 0.6446 | 0.7203 | 0.6100 |
| | DrFuse Yao et al. (2024) | 0.6539 | 0.6223 | 0.5466 | 0.7306 | 0.3632 | 0.6610 | 0.7528 | 0.6042 |
| | MERL Liu et al. (2024) | 0.6579 | 0.6261 | 0.5573 | 0.7005 | 0.3639 | 0.6482 | **0.8027** | 0.6152 |
| | **Ours** | **0.7066** | **0.6753** | **0.6156** | **0.8264** | **0.3825** | **0.6901** | 0.7989 | **0.6784** |
| EMG | BSTT Liu & Jia (2023) | 0.3155 | 0.0959 | 0.0000 | 0.0000 | 0.0000 | **0.4797** | 0.0000 | 0.0000 |
| | XSleepNet Phan et al. (2021) | 0.3883 | 0.3636 | 0.2083 | 0.4413 | 0.1315 | 0.3854 | 0.3553 | **0.5044** |
| | DrFuse Yao et al. (2024) | 0.3395 | 0.2528 | 0.1418 | 0.4906 | 0.0697 | 0.3662 | 0.0196 | 0.3181 |
| | MERL Liu et al. (2024) | 0.3786 | 0.3458 | 0.2012 | 0.5066 | **0.1779** | 0.3902 | 0.2308 | 0.4235 |
| | **Ours** | **0.4190** | **0.3705** | **0.2382** | **0.5609** | 0.0748 | 0.4190 | **0.3655** | 0.4323 |

Table 8: Ablation study of feature weights on two public datasets. The **bold** items denote the best results.

| Dataset | Weight | Modality | Overall Results | | |
|---|---|---|---|---|---|
| | | | Accuracy | Macro F1 | Kappa |
| ISRUC-S3 | $-r_i^{u_a}$ | Multimodal | 0.7910 | 0.7755 | 0.7315 |
| | $exp(-r_i^{u_a})$ | | **0.7930** | **0.7815** | **0.7344** |
| | $-r_i^{u_a}$ | EEG | 0.7501 | 0.7259 | 0.6794 |
| | $exp(-r_i^{u_a})$ | | **0.7646** | **0.7397** | **0.6969** |
| | $-r_i^{u_a}$ | EOG | 0.7317 | 0.7087 | 0.6541 |
| | $exp(-r_i^{u_a})$ | | **0.7444** | **0.7168** | **0.6697** |
| | $-r_i^{u_a}$ | EMG | 0.4194 | 0.4018 | 0.2523 |
| | $exp(-r_i^{u_a})$ | | **0.4384** | **0.4075** | **0.2693** |
| MASS-SS3 | $-r_i^{u_a}$ | Multimodal | 0.8683 | **0.8193** | 0.8050 |
| | $exp(-r_i^{u_a})$ | | **0.8686** | **0.8193** | **0.8058** |
| | $-r_i^{u_a}$ | EEG | 0.8474 | 0.7779 | 0.7731 |
| | $exp(-r_i^{u_a})$ | | **0.8517** | **0.7871** | **0.7798** |
| | $-r_i^{u_a}$ | EOG | 0.8195 | 0.7478 | 0.7305 |
| | $exp(-r_i^{u_a})$ | | **0.8227** | **0.7534** | **0.7359** |
| | $-r_i^{u_a}$ | EMG | 0.5350 | 0.3734 | 0.2514 |
| | $exp(-r_i^{u_a})$ | | **0.5408** | **0.3770** | **0.2613** |

A.5 DETAILED ANALYSIS AND PROOFS IN SECTION 4

We seek to establish a robustness bound that quantifies the amount of useful information transferred from auxiliary modalities to the primary modality under perturbations. This robustness is crucial for ensuring reliable learning and decision-making in scenarios where the auxiliary modalities might be noisy or uncertain. Details are as follows:

**Definition**. We introduce the notations and mathematical definitions used in this section:

1) The primary modality is denoted as $x_i^{u_p}$, and the auxiliary modality as $x_i^{u_a}$, where $u_p$ and $u_a$ represent the primary and auxiliary modalities, respectively. These modalities provide complementary sources of information for classification tasks.

2) Feature maps extracted from the primary and auxiliary modalities are denoted as $f_i^{u_p}$ and $f_i^{u_a}$, respectively. These features are assumed to encapsulate modality-specific representations of the input data.

3) The uncertainty weight $w_i^{u_a}$ for the auxiliary modality is defined as:

$$w_i^{u_a} = \exp(-r_i^{u_a}), \tag{11}$$

where $r_i^{u_a}$ is the uncertainty estimate of the auxiliary modality. A lower uncertainty weight ($w_i^{u_a}$) corresponds to higher uncertainty ($r_i^{u_a}$) in the auxiliary modality, thereby reducing its contribution to the overall learning process.

4) The perturbation between the auxiliary and primary modalities is given by:

$$\delta_i = f_i^{u_a} - f_i^{u_p}. \tag{12}$$

This represents the feature-level discrepancy between the two modalities, which plays a key role in quantifying the robustness of information transfer.

5) The classification margin $M_i$ for the primary modality is defined as:

$$M_i = \hat{y}_i^{u_p} - \max_{k \neq y_i}[\hat{y}_i^{u_p}]_k, \tag{13}$$

where $y_i$ is the true category label, and $[\hat{y}_i^{u_p}]_k$ is the logit of the $k$-th category for the primary modality. The margin $M_i$ measures the separation between the score of the true category and the highest score among incorrect categories. A larger margin implies a more confident decision by the classifier.

6) The total information transfer $I_{\text{total}}^{u_a \to u_p}$ from auxiliary modalities $u_a$ to the primary modality $u_p$ is defined as:

$$I_{\text{total}}^{u_a \to u_p} = \sum_{u_a} w_i^{u_a} \cdot \|f_i^{u_p} - f_i^{u_a}\|_2. \tag{14}$$

This term quantifies the cumulative contribution of auxiliary modalities to the primary modality. The term $w_i^{u_a}$ scales the contribution based on uncertainty, while the $L_2$-norm $\|f_i^{u_p} - f_i^{u_a}\|_2$ captures feature-level similarity between modalities.

7) The perturbation radius $R$ is the minimum $L_2$-norm of $\delta_i$ that satisfies the margin regularization constraint:

$$R = \min_{\delta_i} \|\delta_i\|_2 \quad \text{subject to} \quad M_i \geq \max(0, \hat{y}_i^{u_p} - \hat{y}_i^{u_a} - \epsilon), \tag{15}$$

where $\epsilon > 0$ is a small margin parameter. The perturbation radius $R$ provides a quantitative measure of the model's robustness under feature perturbations.

**Theorem 1**. Suppose the auxiliary modality $u_a$ is associated with an uncertainty weight $w_i^{u_a} = \exp(-r_i^{u_a})$, where $r_i^{u_a} \geq 0$ quantifies the uncertainty of $u_a$. Assume the classification margin $M_i$ satisfies the regularization condition:

$$M_i \geq \max(0, \hat{y}_i^{u_p} - \hat{y}_i^{u_a} - \epsilon), \tag{16}$$

where $\epsilon > 0$ is a small positive margin parameter, $\hat{y}_i^{u_p}$ and $\hat{y}_i^{u_a}$ are the logits of the true category predicted by the primary and auxiliary modalities, respectively.

Under this condition, the following results hold:

1) The perturbation radius $R$ satisfies:

$$R = \min_{\delta_i} \|\delta_i\|_2 \quad \text{with} \quad \|\delta_i\|_2 \geq w_i^{u_a} \cdot \|f_i^{u_a} - f_i^{u_p}\|_2. \tag{17}$$

The perturbation radius quantifies the smallest discrepancy between features of the primary and auxiliary modalities that still maintain the margin regularization condition. The scaling by $w_i^{u_a}$ ensures that modalities with higher uncertainty contribute less to the effective perturbation.

2) The total information transfer $I_{\text{total}}^{u_a \rightarrow u_p}$ is bounded by:

$$\sum_{u_a} w_i^{u_a} \cdot \|f_i^{u_p} - f_i^{u_a}\|_2 \leq I_{\text{total}}^{u_a \rightarrow u_p} \leq \sum_{u_a} \|f_i^{u_p} - f_i^{u_a}\|_2. \tag{18}$$

This bound ensures that the total contribution of auxiliary modalities to the primary modality is appropriately constrained by the uncertainty weights $w_i^{u_a}$ and the feature similarities.

3) A minimax optimization ensures the robustness of the primary modality:

$$\min_{\delta_i} \max_{w_i^{u_a}} \|\delta_i\|_2, \tag{19}$$

subject to $M_i^{\text{smooth}} \geq \epsilon$. The minimax formulation addresses the worst-case scenario by minimizing the maximum possible perturbation under the uncertainty constraints. This strategy ensures that the model remains robust even when auxiliary modalities have high levels of uncertainty.

**Proofs of Theorem 1:**

1) *Perturbation Radius and Margin Regularization:* The perturbation radius $R$ is defined as:

$$R = \min_{\delta_i} \|\delta_i\|_2 \quad \text{subject to} \quad M_i \geq \max(0, \hat{y}_i^{u_p} - \hat{y}_i^{u_a} - \epsilon). \tag{20}$$

Incorporating uncertainty weights $w_i^{u_a}$, the scaled perturbation satisfies:

$$\|\delta_i\|_2 \geq w_i^{u_a} \cdot \|f_i^{u_a} - f_i^{u_p}\|_2. \tag{21}$$

2) *Smoothing Process:* To account for random fluctuations, the smoothed margin $M_i^{\text{smooth}}$ is defined as:

$$M_i^{\text{smooth}} = \frac{1}{T} \sum_{t=1}^{T} M_i(t), \tag{22}$$

where $T$ is the number of perturbations. This smoothing process ensures the model's robustness across multiple perturbation instances.

3) *Bounds on Information Transfer:* Using the definition of total information transfer:

$$I_{\text{total}}^{u_a \rightarrow u_p} = \sum_{u_a} w_i^{u_a} \cdot \|f_i^{u_p} - f_i^{u_a}\|_2. \tag{23}$$

The bounds are established by considering the extreme cases of maximum and minimum uncertainty weights:

$$\sum_{u_a} w_i^{u_a} \cdot \|f_i^{u_p} - f_i^{u_a}\|_2 \leq I_{\text{total}}^{u_a \rightarrow u_p} \leq \sum_{u_a} \|f_i^{u_p} - f_i^{u_a}\|_2. \tag{24}$$

4) *Minimax Optimization:* To ensure robustness under worst-case conditions:

$$\min_{\delta_i} \max_{w_i^{u_a}} \|\delta_i\|_2, \tag{25}$$

subject to $M_i^{\text{smooth}} \geq \epsilon$. This ensures the robustness of the primary modality by minimizing the impact of high-uncertainty auxiliary modalities on the optimization objective.

This concludes the proof.

## A.6 DISCUSSION AND LIMITATIONS

Our limitations include that our method simulates missing signals to assess auxiliary modality quality, but it does not address cases where primary or auxiliary modalities are actually missing in real-world scenarios. Additionally, we did not conduct experiments on datasets with ubiquitous modalities (e.g., ear-EEG). We plan to address these limitations in future work.

In addition, we sincerely appreciate reviewer 'Dg5B' for their valuable suggestions. Indeed, our method has the potential to be applied to other multimodal fields and achieve meaningful benefits, and we plan to explore this further in the future. We also extend our gratitude to reviewer 'A2VX' for their insightful feedback. In this study, we primarily focus on single-modality sleep staging experiments to ensure a more comfortable and practical approach for ubiquitous scenarios. Moving forward, we aim to investigate additional combinations of modalities to enhance our methodology further.

