# OpenReview forum: "SleepSMC: Ubiquitous Sleep Staging via Supervised Multimodal Coordination"
_ICLR.cc/2025/Conference — ICLR 2025 Poster_

### Official Review · Reviewer_A2VX · 2024-10-30

**Soundness:** 3
**Presentation:** 3
**Contribution:** 2
**Rating:** 6
**Confidence:** 4

**Summary:**

The manuscript proposes SleepSMC, a method for detecting sleep stages leveraging contrastive learning and feature weighting based on uncertainty. The approach is specifically designed to handle scenarios where multiple data modalities are available during training, but only a single modality is accessible during testing (multimodal scenario has also been evaluated). Using three publicly available datasets, the authors demonstrate that SleepSMC achieves performance improvements over existing baselines.

**Strengths:**

- The paper is well-structured and clear, and easy to follow.
- Includes extensive experiments comparing the proposed method with various baselines.

**Weaknesses:**

- Limited technical novelty.
- More recent & missing-modality-specific baselines could be considered as stronger baselines.

[Details]
1. Could the authors explain why "Supervised Contrastive Learning" by Khosla et al. (2020) was not considered as one of the baselines? What specific motivations underlie the use of contrastive learning here, and in what ways does the proposed technique differ (in terms of technical novelty) from the original approach in Khosla et al.?
        - Khosla, Prannay, et al. "Supervised contrastive learning." Advances in Neural Information Processing Systems 33 (2020): 18661-18673.

2. Is there any method or technique in this work that is specifically tailored for sleep stage detection? The proposed approach appears applicable to most multimodal data; If that's the case, other recent works targeting learning/inference with missing modalities could be considered as baselines as well:
    - Wang, Hu, et al. "Multi-modal learning with missing modality via shared-specific feature modelling." Proceedings of the IEEE/CVF Conference on Computer Vision and Pattern Recognition. 2023.
   - Yao, Wenfang, et al. "DrFuse: Learning Disentangled Representation for Clinical Multi-Modal Fusion with Missing Modality and Modal Inconsistency." Proceedings of the AAAI Conference on Artificial Intelligence. Vol. 38. No. 15. 2024.

**Questions:**

[Problem setting & method]
1. Could the authors clarify the rationale for assuming that only one modality is available during inference? While it is understandable that not all modalities used in training may be accessible at test time, it seems uncommon to assume the presence of only a single modality. A more practical scenario might involve each user having a unique set of available modalities at test time, rather than relying on one fixed modality. Could the authors elaborate on the motivation behind this specific setting?

2. Would it be expected that the auxiliary modality classifiers exhibit higher uncertainty at the beginning of training? If so, could this uncertainty indicate an opportunity for the model to learn more effectively? Did the authors experiment with dynamic tuning of weights over the course of training? Additionally, numerous metrics (such as entropy, model confidence, loss) could potentially assess sample importance. What was the rationale for selecting model confidence? Were other metrics considered, and if so, what were the comparative results?

[Related work & Baselines]
1. In Section 2, Liu et al. (2023b) and Liu et al. (2024) are referenced as leveraging multimodal data to enhance performance, with the former employing contrastive learning to align modalities. Could the authors clarify why these methods were not considered as baselines? Additionally, it is noted that these works focus on multimodal consistency without capturing class-specific information. However, the design choices of related works should not necessarily be considered limitations. Could the authors discuss the impact of excluding class-specific information? How can we assess the relative performance without a comparison to the proposed method?

2.  In Section 2, several works related to uncertainty estimation are discussed, but the conclusion remains somewhat ambiguous. Why are these methods not applicable to the target scenario? What challenges prevent their direct application to the proposed task, especially given that multimodal data are still used during training?

[Results]
1. In most scenarios, including unimodal cases, the primary improvements appear to stem from supervised contrastive learning. However, in Section 5.5, it is stated that “uncertainty-based feature weighting has a greater impact in the unimodal testing scenario,” which may be an overstatement. Could the authors provide comparative improvement results (e.g., average ± standard deviation) for both components? If the enhancement from uncertainty-based feature weighting is minor, what justifies its inclusion?

2. Could the authors explain the observed fluctuations in EEG and EOG uncertainty weights as training progresses (as seen in Figure 3c)?

[Minor comments]:
- Font size in Figure 3 appears small.
- Class labels in Figure 4 are difficult to read; increasing the legend font size could improve readability.
- Typo in Section 4.2, first sentence: “adaptively weighted” should be “adaptively weights.”

---

> ### Author Response · Authors · 2024-11-20
> **Response to [A2VX] (Part 1/n)**
>
> Thank you for recognizing and supporting the writing and experimental work in our paper! We understand that your main concerns are related to methodological innovation and some experimental details. Below, we address these points:
>
> 1. **Why was "Supervised Contrastive Learning" not used as a baseline? What is the motivation for contrastive learning?**
>
>    The "Supervised Contrastive Learning" method is built on contrastive learning between augmented and original data samples, aiming to enhance general representation capabilities. In contrast, our contrastive learning approach focuses more on inter-modal coordination and alignment, with the goal of improving single-modality robustness and performance in an end-to-end manner.
>
>    To evaluate the effect of data augmentation and provide a meaningful comparison, we supplemented our experiments with the recent method you suggested: MERL (Liu et al., ICML2024). This method incorporates both inter-modal and intra-modal contrastive learning. Specifically, inter-modal alignment facilitates information transfer, while intra-modal contrastive learning leverages data augmentation to enhance representation and robustness. From the experimental results, our method still demonstrates significant superiority.
>
> 2. **Should methods for handling missing modalities in other fields also be compared?**
>
>    Thank you for your suggestion. ShaSpec (CVPR 2023) and DrFuse (AAAI 2024) are excellent works, and we will cite and compare them in the revised paper's related work section. Additionally, we supplemented comparisons with DrFuse (AAAI 2024) under all scenarios across three datasets. The experimental results show that our method still outperforms these approaches.(The table has been combined with Question 5 below.)
>
> 3. **What is the significance of single-modality available? Why not consider a set of modalities?**
>
>    In sleep monitoring, multimodal devices are often complex, difficult to wear, and significantly disrupt sleep experiences, posing substantial obstacles to the monitoring process. Such devices are not suitable for ubiquitous scenarios (e.g., home settings) and may even decrease monitoring accuracy. We are developing a single-modality device aimed at enabling anyone, even without specialized medical knowledge, to perform comfortable, convenient, and accurate sleep monitoring through simple contact.
>
>    Therefore, our method primarily focuses on single-modality testing scenarios. Thank you for pointing this out. We have added this limitation in the "Discussion and Limitation" section of the revised paper.

---

> ### Author Response · Authors · 2024-11-20
> **Response to [A2VX] (Part 2/n)**
>
> 4. **Do you expect high uncertainty at the beginning of training? Any experiments on dynamic weight adjustment? Experiments with alternative metrics?**
>
>    1) In fact, we do not expect high uncertainty, as it is undesirable. Ideally, uncertainty should be low for perfect modality coordination learning. Our goal is to enable the model to overcome situations of high uncertainty and large differences, thereby improving robustness.
>
>    2) All weights in our method are dynamically computed and adjusted, adapting at each step during training.
>
>    3) We conducted experiments using the uncertainty variance metric, directly applying the negative value of formula (5), $-r_i^{u_a}$. However, as explained in formula (6), *"the exponential function introduces a smooth, continuous inverse scaling, where weights decrease more sharply for high uncertainty and more gradually for low uncertainty, reflecting the varying tolerance of the model."* This approach achieves better performance. Details are shown in below:
>
> | Dataset   | Metric                  | Modality    | Accuracy  | Macro F1  | Kappa   |
> |-----------|-------------------------|-------------|-----------|-----------|---------|
> | ISRUC-S3  | $ -r_i^{u_a} $   | Multimodal  | 0.7910    | 0.7755    | 0.7315  |
> |           | $ \exp(-r_i^{u_a}) $ | Multimodal  | **0.7930** | **0.7815** | **0.7344** |
> |           | $ -r_i^{u_a} $   | EEG         | 0.7501    | 0.7259    | 0.6794  |
> |           | $ \exp(-r_i^{u_a}) $ | EEG         | **0.7646** | **0.7397** | **0.6969** |
> |           | $ -r_i^{u_a} $   | EOG         | 0.7317    | 0.7087    | 0.6541  |
> |           | $ \exp(-r_i^{u_a}) $ | EOG         | **0.7444** | **0.7168** | **0.6697** |
> |           | $ -r_i^{u_a} $   | EMG         | 0.4194    | 0.4018    | 0.2523  |
> |           | $ \exp(-r_i^{u_a}) $ | EMG         | **0.4384** | **0.4075** | **0.2693** |
> | MASS-SS3  | $ -r_i^{u_a} $   | Multimodal  | 0.8683    | 0.8193    | 0.8050  |
> |           | $ \exp(-r_i^{u_a}) $ | Multimodal  | **0.8686** | **0.8193** | **0.8058** |
> |           | $ -r_i^{u_a} $   | EEG         | 0.8474    | 0.7779    | 0.7731  |
> |           | $ \exp(-r_i^{u_a}) $ | EEG         | **0.8517** | **0.7871** | **0.7798** |
> |           | $ -r_i^{u_a} $   | EOG         | 0.8195    | 0.7478    | 0.7305  |
> |           | $ \exp(-r_i^{u_a}) $ | EOG         | **0.8227** | **0.7534** | **0.7359** |
> |           | $ -r_i^{u_a} $   | EMG         | 0.5350    | 0.3734    | 0.2514  |
> |           | $ \exp(-r_i^{u_a}) $ | EMG         | **0.5408** | **0.3770** | **0.2613** |

---

> ### Author Response · Authors · 2024-11-20
> **Response to [A2VX] (Part 3/n)**
>
> 5. **Can unsupervised contrastive learning methods mentioned in the paper be fairly compared?**
>
>    Thank you for your suggestion. We supplemented our comparisons with the method by Liu et al. (2024) across all scenarios on three datasets. To ensure a fair comparison, we adapted the method into a supervised, end-to-end approach while retaining its core modality-level and intra-modal contrastive learning components.
>
> - Multimodal scenario:
>
> | Dataset       | Method        | Accuracy  | Macro F1  | Kappa   |
> |---------------|---------------|-----------|-----------|---------|
> | **ISRUC-S3**  | BSTT          | 0.7756    | 0.7568    | 0.7114  |
> |               | XSleepNet     | 0.6705    | 0.6440    | 0.5771  |
> |       | **DrFuse**        | 0.7741    | 0.7469    | 0.7091  |
> |               | **MERL**          | 0.7559    | 0.7458    | 0.6876  |
> |               | **Ours**      | **0.7930** | **0.7815** | **0.7344** |
> |  **MASS-SS3** | BSTT          | 0.8114    | 0.7492    | 0.7190  |
> |               | XSleepNet     | 0.8066    | 0.7464    | 0.7158  |
> |      | **DrFuse**        | 0.8628    | 0.8086    | 0.7964  |
> |               | **MERL**          | 0.8605    | 0.8055    | 0.7915  |
> |               | **Ours**      | **0.8686** | **0.8193** | **0.8058** |
> | **Sleep-EDF-78** | BSTT       | 0.7321    | 0.6335    | 0.6245  |
> |               | XSleepNet     | 0.7577    | 0.6855    | 0.6631  |
> |   | **DrFuse**        | 0.8009    | 0.7411    | 0.7235  |
> |               | **MERL**          | 0.7990    | 0.7267    | 0.7196  |
> |               | **Ours**      | **0.8158** | **0.7558** | **0.7450** |
>
> - Unimodal scenario + ISRUC-S3 dataset:
>
> | Dataset       | Method        | Accuracy  | Macro F1  | Kappa   |
> |---------------|---------------|-----------|-----------|---------|
> |    **EEG**      | BSTT          | 0.7191    | 0.6921    | 0.6371  |
> |               | XSleepNet     | 0.6555    | 0.6322    | 0.5614  |
> |       | **DrFuse**        | 0.7532    | 0.7138    | 0.6818  |
> |               | **MERL**          | 0.7467    | 0.7295    | 0.6758  |
> |               | **Ours**      | **0.7646** | **0.7397** | **0.6969** |
> |    **EOG**      | BSTT          | 0.4700    | 0.3163    | 0.2790  |
> |               | XSleepNet     | 0.6288    | 0.6071    | 0.5233  |
> |         | **DrFuse**        | 0.6947    | 0.6799    | 0.6078  |
> |               | **MERL**          | 0.6976    | 0.6741    | 0.6132  |
> |               | **Ours**      | **0.7444** | **0.7168** | **0.6697** |
> |    **EMG**       | BSTT          | 0.3046    | 0.0934    | 0.0000  |
> |               | XSleepNet     | 0.3660    | 0.3484    | 0.1935  |
> |       | **DrFuse**        | 0.3857    | 0.3789    | 0.2318  |
> |               | **MERL**          | 0.3981    | 0.3907    | 0.2348  |
> |               | **Ours**      | **0.4384** | **0.4075** | **0.2693** |
>
> - Unimodal scenario + the other datasets:
> Please refer to Section A.3 of the revised manuscript for the full results and analysis.
>
> **Reference** \
> [1] Liu Y, Jia Z. Bstt: A bayesian spatial-temporal transformer for sleep staging[C]//The Eleventh International Conference on Learning Representations. 2023.\
> [2] Phan H, Chén O Y, Tran M C, et al. XSleepNet: Multi-view sequential model for automatic sleep staging[J]. IEEE Transactions on Pattern Analysis and Machine Intelligence, 2021, 44(9): 5903-5915.\
> [3] Yao W, Yin K, Cheung W K, et al. DrFuse: Learning Disentangled Representation for Clinical Multi-Modal Fusion with Missing Modality and Modal Inconsistency[C]//Proceedings of the AAAI Conference on Artificial Intelligence. 2024, 38(15): 16416-16424.\
> [4] Liu C, Wan Z, Ouyang C, et al. Zero-Shot ECG Classification with Multimodal Learning and Test-time Clinical Knowledge Enhancement[C]//Forty-first International Conference on Machine Learning, 2024.

---

> ### Author Response · Authors · 2024-11-20
> **Response to [A2VX] (Part 4 /n)**
>
> 6. **Why are uncertainty estimation methods mentioned in related work not applicable to the target scenario?**
>
>    Thank you for pointing this out. We will elaborate on this in the revised manuscript. These methods focus on effective multimodal fusion, including retaining useful modality information and removing irrelevant data. However, they do not address single-modality application scenarios, which are our primary concern. Their methods require all modalities present during the testing phase, and retaining only one modality significantly degrades accuracy. In contrast, our method improves the performance of individual modalities through coordinated learning, enabling robust single-modality applications in ubiquitous scenarios.
>
> 7. **The performance improvement of the uncertainty-based weighting module is smaller than that of the contrastive learning module. How is randomness mitigated?**
>
>    Thank you for highlighting this. To mitigate randomness, we have taken the following steps:
>
>    - **Experimentation**: As described in Section 5.2 of the original paper, we conducted extensive experiments, and all reported results are aggregated from five-fold cross-validation.
>
>    - **Metrics**: To account for the impact of randomness, we introduced the Kappa metric. Kappa is a robust metric designed to measure agreement between predictions and ground truth while accounting for agreement due to chance. The Kappa results clearly illustrate the contribution of the uncertainty-based weighting module.
>
>    - **Writing**: Thank you for pointing this out. We acknowledge that the role of the uncertainty-based weighting module is smaller than that of the contrastive learning module. Our original language may have been misleading, implying that most of the performance gain came from the uncertainty-based module. We have revised this statement to: *"The Supervised Modality-level Contrastive Coordination module plays a more significant role in both scenarios. Meanwhile, the Uncertainty-based Feature Weighting module demonstrates relatively enhanced performance in unimodal compared to multimodal scenarios."* Thank you again for your thoughtful feedback, which has improved our paper.
>
> 8. **Fluctuations in Figure 3c?**
>
>    The robustness gap between EMG and the other two modalities is significant, resulting in less fluctuation for EMG compared to EEG and EOG. For EEG and EOG, their robustness is closer, leading to relative changes in their rankings. Additionally, robustness gaps always exist, and our goal is not to eliminate them but to demonstrate the significance of our method through these gaps. Our method dynamically leverages these differences to achieve better information transfer.
>
> 9. **Minor writing suggestions?**
>
>    Thank you for your meticulous review, which has made our paper better! We have incorporated your suggestions into the revised manuscript and have thoroughly checked the entire paper for any writing issues.

---

> ### Comment · Reviewer_A2VX · 2024-11-25
>
> The authors have addressed my comments, including comparison with more recent baselines, clarification of the results analysis, and the experiment setting. I appreciate the authors' efforts in making the necessary adjustments. I believe the quality of paper has improved, and I have increased my score accordingly.

---

> ### Author Response · Authors · 2024-11-27
> **Thanks for Reviewer [A2VX]**
>
> Dear reviewer,
>
> Thank you for your time and effort in reviewing our paper, and thank you for your recognition and positive feedback on our paper. Your support has brought us great encouragement.
>
> We are really happy to address all your concerns, and we are very grateful for your help and encouragement, which makes our paper better and better. If you have further questions, we will be happy to receive your feedback.
>
> Thank you again for your support and valuable insights!
>
> Best wishes

---

### Official Review · Reviewer_HHyg · 2024-10-31

**Soundness:** 2
**Presentation:** 3
**Contribution:** 2
**Rating:** 5
**Confidence:** 5

**Summary:**

To address the issue that only one modality is available in ubiquitous scenarios, this paper introduce multimodal collaboration in sleep staging, leveraging multiple auxiliary modalities to improve the performance of primary modality-based sleep staging in an end-to-end manner. The paper utilize supervised modality-level instance contrastive coordination and uncertainty estimates to learn coordinated features. The experiment results show that the proposed method achieves SOTA performance in multimodal scenarios and unimodal scenarios.

**Strengths:**

The paper aims to address the issue that only one modality is available in ubiquitous scenarios, which has certain practical significance.

The paper utilizes uncertainty estimates to adaptively weight auxiliary modality features during
training, which ensures that more reliable auxiliary modality features contribute more significantly to the contrastive learning process.

The presentation of the paper is generally quite clear.

**Weaknesses:**

(1) This paper does not introduce a new concept; rather, it applies existing multimodal coordination and uncertainty estimates to the ubiquitous scenarios of sleep staging. Moreover, the proposed method for improving sleep staging does not seem to differ significantly from the multimodal coordination methods already used in areas such as vision. Some design details in the method, such as certain aspects related to uncertainty estimates, do not demonstrate a specific focus on the sleep staging task.

(2) The authors seem to use a single-epoch sleep staging paradigm instead of a sequence-to-sequence sleep staging approach. Why is this the case? In fact, there is a strong correlation between adjacent epochs in sleep staging, and the single-epoch paradigm has already fallen behind in performance and been largely abandoned. As a paper aimed at improving sleep staging performance, it is difficult to understand the choice of using a single-epoch paradigm. Could the authors explain why this paradigm was chosen and whether the proposed method can be adapted to the sequence-to-sequence paradigm?

(3) The reviewer has some concerns about the experimental design and results. The baselines chosen by the authors appear to be relatively weak and outdated, such as the selection of SimCLR, which is a very old self-supervised learning method. The baselines in the paper perform very weakly in multimodal scenarios (shown in Table 1), making it difficult to demonstrate the advantages of the proposed method. Some very strong baselines, such as SalientSleepNet [1], BSTT [2] and XSleepNet [3] were not compared. Meanwhile, the reviewer cannot understand why the authors chose ISRUC-S3 instead of ISRUC-S1 as the primary evaluation dataset. ISRUC-S1 has more subjects and a larger data volume, making it more convincing compared to ISRUC-S3.

(4) Although the presentation of the paper is relatively clear, the content in "DETAILED ANALYSIS AND PROOFS IN SECTION 4" is difficult for readers to understand. As a supplement and proof of certain concepts in the main text, this section does not sufficiently clarify the formal definitions of each concept and the complete proof process. For example, what is the complete proposition that needs to be proven in this section? What are the assumptions underlying the proof process? What is the formal definition of "margin"? What is the formal definition of "information transfer"? The authors should provide clear explanations and complete definitions of these concepts in the appendix to ensure the readability of the paper.

[1] Jia Z, Lin Y, Wang J, et al. SalientSleepNet: Multimodal salient wave detection network for sleep staging[J]. arXiv preprint arXiv:2105.13864, 2021.

[2] Liu Y, Jia Z. Bstt: A bayesian spatial-temporal transformer for sleep staging[C]//The Eleventh International Conference on Learning Representations. 2023.

[3] Phan H, Chén O Y, Tran M C, et al. XSleepNet: Multi-view sequential model for automatic sleep staging[J]. IEEE Transactions on Pattern Analysis and Machine Intelligence, 2021, 44(9): 5903-5915.

**Questions:**

Please See Weakness

---

> ### Author Response · Authors · 2024-11-20
> **Response to [HHyg] (Part 1/n)**
>
> Thank you for recognizing and supporting the practical significance and writing quality of our paper! We understand that your main concerns are related to the methodological innovation and some experimental details. Below, we address these points:
>
> 1. **This paper does not introduce a new concept, but only combines and applies existing methods?**
>
>    Our method incorporates supervised modality-level contrastive coordination and uncertainty-based weighting. The former introduces label information during modality alignment to enhance accuracy, aligning not only modality instances at the same time but also across different times, fully learning the category-relevant temporal information. The latter facilitates dynamic filtering of information during transmission, significantly improving the model's robustness.
>
>    The combination of these two innovations effectively achieves single-modality sleep staging in ubiquitous scenarios. Therefore, while our work demonstrates innovation and practical significance in its application, it also includes substantial methodological novelty.
>
> 2. **Why not adopt a sequence-to-sequence paradigm? Can this method adapt to a sequence-to-sequence paradigm?**
>
>    1) Many existing methods are not based on a sequence-to-sequence paradigm but still achieve excellent results. For example, several recent methods:
>
>         [1] Jia Z, Wang H, Liu Y, et al. Mutual Distillation Extracting Spatial-temporal Knowledge for Lightweight Multi-channel Sleep Stage Classification[C]//Proceedings of the 30th ACM SIGKDD Conference on Knowledge Discovery and Data Mining. 2024: 1279-1289.\
>         [2] Thapa R, He B, Kjaer M R, et al. SleepFM: Multi-modal Representation Learning for Sleep Across Brain Activity, ECG and Respiratory Signals[C]//Forty-first International Conference on Machine Learning. 2024.\
>         [3] Zhu H, Zhou W, Fu C, et al. Masksleepnet: A cross-modality adaptation neural network for heterogeneous signals processing in sleep staging[J]. IEEE Journal of Biomedical and Health Informatics, 2023, 27(5): 2353-2364.
>
>     2) Although our method is not specifically designed as a sequence-to-sequence paradigm, it inherently learns sequence-level information during the supervised contrastive coordination process. Our approach not only aligns sequences at the same time point but also sequences of the same class across different time points. This approach considers sequence-level contextual consistency and effectively utilizes class label information to handle situations such as abrupt class transitions in sleep staging tasks.
>
>         For instance, during transitions between two sleep stages, traditional sequence-to-sequence methods might mistakenly classify adjacent sequences as belonging to the same stage. Our method addresses this issue effectively. Clearly, our approach is adaptable to a sequence-to-sequence paradigm.

---

> ### Author Response · Authors · 2024-11-20
> **Response to [HHyg] (Part 2/n)**
>
> 3. **The selected baselines are outdated. Why choose ISRUC-S3 instead of S1?**
>
>    Thank you for your suggestion. The three methods you mentioned are highly influential in the sleep staging domain. We will include these three papers in the revised manuscript's related work section and provide comparisons. Additionally, we supplemented the experiments with five recent methods (including BSTT and XSleepNet), and the results under two scenarios demonstrate that our method still achieves the best performance.
>
>    Regarding our choice of the smaller S3 subset instead of S1, there are two main considerations:
>    - **Dataset scale**: Although the SleepEDF dataset only includes 78 subjects, its total data volume is much larger than that of S1 and is often considered a large-scale dataset. We have already validated our method on SleepEDF.
>    - **Scenario**: S3 and S1 originate from the same source. Our method focuses on multimodal robustness in ubiquitous and low-resource scenarios. Thus, we prioritized validating performance on low-resource, small-scale datasets like S3.
>
> - Multimodal scenario:
>
> | Dataset       | Method        | Accuracy  | Macro F1  | Kappa   |
> |---------------|---------------|-----------|-----------|---------|
> | **ISRUC-S3**  | **BSTT**          | 0.7756    | 0.7568    | 0.7114  |
> |               | **XSleepNet**     | 0.6705    | 0.6440    | 0.5771  |
> |               | DrFuse        | 0.7741    | 0.7469    | 0.7091  |
> |               | MERL          | 0.7559    | 0.7458    | 0.6876  |
> |               | **Ours**      | **0.7930** | **0.7815** | **0.7344** |
> | **MASS-SS3**  | **BSTT**          | 0.8114    | 0.7492    | 0.7190  |
> |               | **XSleepNet**     | 0.8066    | 0.7464    | 0.7158  |
> |               | DrFuse        | 0.8628    | 0.8086    | 0.7964  |
> |               | MERL          | 0.8605    | 0.8055    | 0.7915  |
> |               | **Ours**      | **0.8686** | **0.8193** | **0.8058** |
> | **Sleep-EDF-78** | **BSTT**       | 0.7321    | 0.6335    | 0.6245  |
> |               | **XSleepNet**     | 0.7577    | 0.6855    | 0.6631  |
> |               | DrFuse        | 0.8009    | 0.7411    | 0.7235  |
> |               | MERL          | 0.7990    | 0.7267    | 0.7196  |
> |               | **Ours**      | **0.8158** | **0.7558** | **0.7450** |
>
>
> - Unimodal scenario + ISRUC-S3 dataset:
>
> | Modality       | Method        | Accuracy  | Macro F1  | Kappa   |
> |---------------|---------------|-----------|-----------|---------|
> |    **EEG**    | **BSTT**          | 0.7191    | 0.6921    | 0.6371  |
> |               | **XSleepNet**     | 0.6555    | 0.6322    | 0.5614  |
> |               | DrFuse        | 0.7532    | 0.7138    | 0.6818  |
> |               | MERL          | 0.7467    | 0.7295    | 0.6758  |
> |               | **Ours**      | **0.7646** | **0.7397** | **0.6969** |
> |    **EOG**    | **BSTT**          | 0.4700    | 0.3163    | 0.2790  |
> |               | **XSleepNet**     | 0.6288    | 0.6071    | 0.5233  |
> |               | DrFuse        | 0.6947    | 0.6799    | 0.6078  |
> |               | MERL          | 0.6976    | 0.6741    | 0.6132  |
> |               | **Ours**      | **0.7444** | **0.7168** | **0.6697** |
> |   **EMG**     | **BSTT**          | 0.3046    | 0.0934    | 0.0000  |
> |               | **XSleepNet**     | 0.3660    | 0.3484    | 0.1935  |
> |               | DrFuse        | 0.3857    | 0.3789    | 0.2318  |
> |               | MERL          | 0.3981    | 0.3907    | 0.2348  |
> |               | **Ours**      | **0.4384** | **0.4075** | **0.2693** |
>
> - Unimodal scenario + the other datasets:
> Please refer to Section A.3 of the revised manuscript for the full results and analysis.
>
> **Reference** \
> [1] Liu Y, Jia Z. Bstt: A bayesian spatial-temporal transformer for sleep staging[C]//The Eleventh International Conference on Learning Representations. 2023.\
> [2] Phan H, Chén O Y, Tran M C, et al. XSleepNet: Multi-view sequential model for automatic sleep staging[J]. IEEE Transactions on Pattern Analysis and Machine Intelligence, 2021, 44(9): 5903-5915.\
> [3] Yao W, Yin K, Cheung W K, et al. DrFuse: Learning Disentangled Representation for Clinical Multi-Modal Fusion with Missing Modality and Modal Inconsistency[C]//Proceedings of the AAAI Conference on Artificial Intelligence. 2024, 38(15): 16416-16424.\
> [4] Liu C, Wan Z, Ouyang C, et al. Zero-Shot ECG Classification with Multimodal Learning and Test-time Clinical Knowledge Enhancement[C]//Forty-first International Conference on Machine Learning, 2024.
>
> 4. **Supplement and proof not detailed enough?**
>
>    Thank you for your feedback, which has improved the readability of our paper. In the revised version, we have further supplemented the relevant definitions and proofs for greater clarity and completeness.

---

> > ### Comment · Reviewer_HHyg · 2024-11-25
> > **the advantages of the proposed method are not significant.**
> >
> > In the multimodal scenario, compared to other methods, the advantages of the proposed method are not significant (e.g., 0.8605 v.s. 0.8686; 0.8009 v.s. 0.8158). Besides, the S1 also has multiple modality data. I still think S1 is a better choice for the evaluation.

---

> ### Author Response · Authors · 2024-11-27
> **Response to [HHyg] (Part 3/n): Supplemented the ISRUC-S1 dataset**
>
> Dear reviewer:
>
> According to your suggestion, we added experiments on the **ISRUC-S1** dataset with four comparison methods in two scenarios. Thank you for your suggestion, which make our paper better. If you have any additional suggestions or feedback, we would be truly grateful to hear them.
>
> 1. Multimodal scenario:
>
> | Dataset       | Method        | Accuracy  | Macro F1  | Kappa   |
> |---------------|---------------|-----------|-----------|---------|
> | **ISRUC-S1**  | **BSTT**          | 0.7247    | 0.6890    | 0.6423  |
> |               | **XSleepNet**     | 0.7444    | 0.7226    | 0.6707  |
> |               | DrFuse        | 0.7441    | 0.7215    | 0.6669  |
> |               | MERL          | 0.7245    | 0.7042    | 0.6417  |
> |               | **Ours**      | **0.7710** | **0.7462** | **0.7018** |
>
>
> 2. Unimodal scenario + **ISRUC-S1** dataset:
>
> | Modality       | Method        | Accuracy  | Macro F1  | Kappa   |
> |---------------|---------------|-----------|-----------|---------|
> |    **EEG**    | **BSTT**          | 0.6840    | 0.6367    | 0.5878  |
> |               | **XSleepNet**     | 0.7092    | 0.6735    | 0.6233  |
> |               | DrFuse        | 0.6978    | 0.6620    | 0.6087  |
> |               | MERL          | 0.7096    | 0.6690    | 0.6223  |
> |               | **Ours**      | **0.7328** | **0.6959** | **0.6536** |
> |    **EOG**    | **BSTT**          | 0.3123    | 0.1115    | 0.0019  |
> |               | **XSleepNet**     | 0.6320    | 0.6047    | 0.5229  |
> |               | DrFuse        | 0.6539    | 0.6223    | 0.5466  |
> |               | MERL          | 0.6579    | 0.6261    | 0.5573  |
> |               | **Ours**      | **0.7066** | **0.6753** | **0.6156** |
> |   **EMG**     | **BSTT**          | 0.3155    | 0.0959    | 0.0000  |
> |               | **XSleepNet**     | 0.3883    | 0.3636    | 0.2083  |
> |               | DrFuse        | 0.3395    | 0.2528    | 0.1418  |
> |               | MERL          | 0.3786    | 0.3458    | 0.2012  |
> |               | **Ours**      | **0.4190** | **0.3705** | **0.2382** |
>
> Best wishes

---

> > ### Comment · Reviewer_HHyg · 2024-11-27
> > **The results are not so convincing. The results of BSTT on both S1 and MASS-S3 reported by the authors are much lower than those reported in the original paper [1].**
> >
> > Thank you for the effort in adding new experiments. I am curious about why the results of BSTT on both S1 and MASS-S3 are significantly lower than those reported in the original paper[1]. For instance, the BSTT result on S1 reported by the authors is 0.7247, while the original paper [1] states it as 0.8196. Why is there such a large discrepancy (0.7247 vs. 0.8196)? Similarly, the BSTT result on MASS-S3 provided by the authors is 0.8114, compared to 0.8950 in the original paper [1]. This gap is also quite substantial.
> >
> > Given that the dataset and task are identical, I believe it would be more reasonable to directly compare the results with those reported in the original paper rather than re-implementing the method. This is why I find the results unconvincing.
> >
> > [1] Liu Y, Jia Z. Bstt: A bayesian spatial-temporal transformer for sleep staging[C]//The Eleventh International Conference on Learning Representations. 2023.

---

> ### Author Response · Authors · 2024-11-27
> **Reason for Not Directly Using the Experimental Results from the Original Paper: Differences in Experimental Setting**
>
> We sincerely apologize for any confusion caused. In fact, our experimental setup differs significantly from that of the original paper. The primary reason for this is that the experimental setting in the original paper [1] is not entirely standardized.
>
> From the publicly available code of the original paper [1], it can be observed that the authors only divided the dataset into a training set and a test set. Specifically, they trained the model on the training set and reported the test set results for the best-performing epoch. Similarly, in another work, MSTGCN [2], the same author of [1] used a comparable setting, as seen in their released code.
>
> However, from a methodological perspective, this approach is not strictly standardized. To better avoid test set data leakage and evaluate the cross-subject generalization performance of the model, we introduced a validation set by randomly splitting 20% of the training set. The model was then trained using the remaining 80% of the training data, and the best-performing model on the validation set was saved for evaluation on the test set.
>
> This approach inevitably reduced the amount of data available for training (to 80%) and ensured that the test set was not directly used for model selection. To maintain fairness, we applied this standardized experimental setup consistently across all our experiments.
>
> This explains why we did not directly cite the results reported in the original paper. We hope this addresses your concerns.
>
>  [1] Liu Y, Jia Z. Bstt: A bayesian spatial-temporal transformer for sleep staging[C]//The Eleventh International Conference on Learning Representations. 2023.
>
> [2] Jia Z, Lin Y, Wang J, et al. Multi-view spatial-temporal graph convolutional networks with domain generalization for sleep stage classification[J]. IEEE Transactions on Neural Systems and Rehabilitation Engineering, 2021, 29: 1977-1986.

---

> ### Author Response · Authors · 2024-12-03
> **Dear Reviewer HHyg**
>
> Dear Reviewer HHyg,
>
> We would like to express our sincere gratitude for your careful review and thoughtful comments, which have greatly helped improve our paper. We deeply appreciate the time and effort you have put into providing such valuable feedback.
>
> As the discussion period is drawing to a close, we humbly hope to earn your support. We have made every effort to address your concerns, and we sincerely welcome any further questions or suggestions you may have. Your insights are truly invaluable to us, and we would be grateful for any additional guidance you can offer.
>
> Thank you again for your time and consideration.

---

> > ### Comment · Reviewer_HHyg · 2024-12-03
> > **Thank you for your efforts and explanation, and I maintain my initial review score.**
> >
> > Thank you for your efforts and explanation. However, I am still confused. It seems unreasonable to have such a significant gap in the results for sleep staging solely due to differences in the experimental settings in the training and validation sets. Therefore, I will maintain my initial review score.

---

### Official Review · Reviewer_iRq4 · 2024-11-02

**Soundness:** 3
**Presentation:** 2
**Contribution:** 2
**Rating:** 6
**Confidence:** 5

**Summary:**

This paper introduced multimodal collaboration in sleep staging, leveraging multiple auxiliary modalities to improve the performance of primary modality-based sleep staging in an end-to-end manner. The authors utilized supervised modality-level instance contrastive coordination to capture category-related consistency and complementarity across intra-modality and inter-modality.

**Strengths:**

Experiments are performed using 3 different datasets. The organization of the paper is good.

**Weaknesses:**

The limitations are as follows:

The motivation is not clear.
How this contribution is different from the following contributions? a)CoRe-Sleep: A Multimodal Fusion Framework for Time Series Robust to Imperfect Modalities," in IEEE Transactions on Neural Systems and Rehabilitation Engineering, vol. 32, pp. 840-849, 2024, doi: 10.1109/TNSRE.2024.3354388. b) Multi-Modal Sleep Stage Classification With Two-Stream Encoder-Decoder," in IEEE Transactions on Neural Systems and Rehabilitation Engineering, vol. 32, pp. 2096-2105, 2024, doi: 10.1109/TNSRE.2024.3394738.
The motivation behind the use of Uncertainty Estimation with Frozen Gradients is not clear.
The methods mentioned in Table 2 are from before 2022. It is necessary to compare with some recent SOTA methods

**Questions:**

1. How the proposed method is different from SOTA?
2. What is the novel contribution that makes this paper unique?
3. What is the computational complexity?
4. How equation 10 become optimized?

---

> ### Author Response · Authors · 2024-11-20
> **Response to [iRq4] (Part 1/n)**
>
> Thank you for recognizing and supporting our experimental work and writing! We understand that your main concerns are related to the motivations behind our method and some implementation details. Below, we address them in detail.
>
> 1. **The motivation is not clear? Where is the novelty of the method?**
>
> Thank you for your recommendation. We have carefully reviewed the CoRe-Sleep and TSEDSleepNet methods you mentioned, which are excellent sleep staging approaches. However, our method is significantly different from theirs, and we will compare and cite it in the revised paper. Details are as follows:
>
>    - **Motivation**: Our method aims to address the problem of multimodal training with single-modality testing, significantly improving single-modality performance during the testing phase. This makes it well-suited for ubiquitous applications, such as home sleep monitoring in ubiquitous scenarios. In contrast, CoRe-Sleep and TSEDSleepNet focus on achieving better multimodal fusion. CoRe-Sleep emphasizes robust fusion, while TSEDSleepNet focuses on class imbalance and temporal modeling capabilities.
>
>    - **Methodological Novelty**: We introduce supervised contrastive learning for modality-level coordinated and associated learning, effectively addressing the inconsistency between modalities during training and testing phases. Additionally, we propose uncertainty-based weighting to facilitate the selection and transmission of high-quality information during coordination. While CoRe-Sleep also performs modality alignment, it relies on unsupervised alignment and does not fully leverage class label information or optimize the alignment process. On the other hand, the primary innovations in TSEDSleepNet lie in its model structure and its loss function designed for class imbalance.

---

> > ### Comment · Reviewer_iRq4 · 2024-11-20
> >
> > In the case of supervised contrastive learning, is it possible to monitor real-time home sleep in ubiquitous scenarios?
> > If yes, then how?

---

> ### Author Response · Authors · 2024-11-20
> **Response to [iRq4] (Part 2/n)**
>
> - **Experiments**: To further demonstrate the effectiveness of our method, we have included comparisons with four state-of-the-art methods, covering approaches for multimodal missing and sleep staging methods. On three datasets, under both multimodal and unimodal scenarios, our method consistently achieves the best performance.
>
> 1. Multimodal scenario:
>
> | Dataset       | Method        | Accuracy  | Macro F1  | Kappa   |
> |---------------|---------------|-----------|-----------|---------|
> | **ISRUC-S3**  | BSTT          | 0.7756    | 0.7568    | 0.7114  |
> |               | XSleepNet     | 0.6705    | 0.6440    | 0.5771  |
> |               | DrFuse        | 0.7741    | 0.7469    | 0.7091  |
> |               | MERL          | 0.7559    | 0.7458    | 0.6876  |
> |               | **Ours**      | **0.7930** | **0.7815** | **0.7344** |
> | **MASS-SS3**  | BSTT          | 0.8114    | 0.7492    | 0.7190  |
> |               | XSleepNet     | 0.8066    | 0.7464    | 0.7158  |
> |               | DrFuse        | 0.8628    | 0.8086    | 0.7964  |
> |               | MERL          | 0.8605    | 0.8055    | 0.7915  |
> |               | **Ours**      | **0.8686** | **0.8193** | **0.8058** |
> | **Sleep-EDF-78** | BSTT       | 0.7321    | 0.6335    | 0.6245  |
> |               | XSleepNet     | 0.7577    | 0.6855    | 0.6631  |
> |               | DrFuse        | 0.8009    | 0.7411    | 0.7235  |
> |               | MERL          | 0.7990    | 0.7267    | 0.7196  |
> |               | **Ours**      | **0.8158** | **0.7558** | **0.7450** |
>
>
> 2. Unimodal scenario + ISRUC-S3 dataset:
>
> | Modality       | Method        | Accuracy  | Macro F1  | Kappa   |
> |---------------|---------------|-----------|-----------|---------|
> |    **EEG**    | BSTT          | 0.7191    | 0.6921    | 0.6371  |
> |               | XSleepNet     | 0.6555    | 0.6322    | 0.5614  |
> |               | DrFuse        | 0.7532    | 0.7138    | 0.6818  |
> |               | MERL          | 0.7467    | 0.7295    | 0.6758  |
> |               | **Ours**      | **0.7646** | **0.7397** | **0.6969** |
> |    **EOG**    | BSTT          | 0.4700    | 0.3163    | 0.2790  |
> |               | XSleepNet     | 0.6288    | 0.6071    | 0.5233  |
> |               | DrFuse        | 0.6947    | 0.6799    | 0.6078  |
> |               | MERL          | 0.6976    | 0.6741    | 0.6132  |
> |               | **Ours**      | **0.7444** | **0.7168** | **0.6697** |
> |   **EMG**     | BSTT          | 0.3046    | 0.0934    | 0.0000  |
> |               | XSleepNet     | 0.3660    | 0.3484    | 0.1935  |
> |               | DrFuse        | 0.3857    | 0.3789    | 0.2318  |
> |               | MERL          | 0.3981    | 0.3907    | 0.2348  |
> |               | **Ours**      | **0.4384** | **0.4075** | **0.2693** |
>
> 3. **Unimodal scenario + the other datasets**:
> Please refer to Section A.3 of the revised manuscript for the full results and analysis.
>
> **Reference** \
> [1] Liu Y, Jia Z. Bstt: A bayesian spatial-temporal transformer for sleep staging[C]//The Eleventh International Conference on Learning Representations. 2023.\
> [2] Phan H, Chén O Y, Tran M C, et al. XSleepNet: Multi-view sequential model for automatic sleep staging[J]. IEEE Transactions on Pattern Analysis and Machine Intelligence, 2021, 44(9): 5903-5915.\
> [3] Yao W, Yin K, Cheung W K, et al. DrFuse: Learning Disentangled Representation for Clinical Multi-Modal Fusion with Missing Modality and Modal Inconsistency[C]//Proceedings of the AAAI Conference on Artificial Intelligence. 2024, 38(15): 16416-16424.\
> [4] Liu C, Wan Z, Ouyang C, et al. Zero-Shot ECG Classification with Multimodal Learning and Test-time Clinical Knowledge Enhancement[C]//Forty-first International Conference on Machine Learning, 2024.

---

> ### Author Response · Authors · 2024-11-20
> **Response for [iRq4] (Part 3/n)**
>
> 2. **What is the computational complexity?**
>
>    Our model uses only a CNN-based structure, its computational complexity is low overall, as detailed in the table below. In particular, the computational complexity during the single-modality testing phase is extremely low, which strongly supports its deployment in ubiquitous single-modality scenarios.
>     | Scenario | Training phase   | Testing phase   |
>     |------------|------------|------------|
>     | Multimodal|33.0M FLOPs & 1.3M param|16.5M FLOPs & 0.6M param|
>     | Unimodal| 26.1M FLOPs & 1.0M param|6.9M FLOPs & 0.2M param|
>
> 3. **How equation 10 become optimized?**
>
>     Equation 10 defines the overall objective function for optimizing SleepSMC. It integrates a supervised contrastive loss $𝐿_{Con}$ to align cross-modal features and a classification loss $𝐿_{cls}$ to ensure accurate sleep stage predictions. The $𝐿_{Con}$ stabilizes the training process by controlling the contribution of auxiliary modalities.
>
>     These losses are jointly optimized through a sum. The joint optimization is performed using stochastic gradient descent (SGD) with the Adam optimizer, which efficiently handles the gradients of both losses.

---

> ### Author Response · Authors · 2024-11-20
> **Response to [iRq4] (Part 4/n): Feasibility of Real-Time Monitoring**
>
> Our method indeed enables real-time sleep monitoring. Within the context of supervised contrastive learning, contrastive learning is performed during the training phase. The goal is to enhance the learning of modality-specific, class-relevant information through multimodal data. This approach significantly improves the model's performance in single-modality testing scenarios. In practical applications, users only need to wear a simple single-modality device, such as an ear-EEG device. The model can process the collected single-modality data to accurately classify sleep stages. By analyzing sleep cycles and structures, effective sleep monitoring becomes straightforward.
>
> **Feasibility of Real-Time Monitoring**:
>
> As discussed in Part 3/n, the computational complexity of single-modality model inference is extremely low, requiring only 6.9M FLOPs. Even on a mid-range ARM CPU (capable of executing 5 GFLOPs per second), the inference time is approximately **1.38 milliseconds**, while a typical clinical sleep segment spans **30 seconds**. Since the inference time is orders of magnitude smaller than the sample duration, real-time monitoring is highly feasible.
>
> Let me know if you have further questions!

---

> ### Author Response · Authors · 2024-11-27
> **Thanks for Reviewer [iRq4]**
>
> Dear reviewer,
>
> Thank you for your time and effort in reviewing our paper, and thank you for your recognition and positive feedback on our paper.
>
> We have provided comprehensive responses to your concerns through detailed explanations and thorough experimental validations. In particular, we have explained the real-time monitoring issue you mentioned.
>
> We hope that we have addressed your concerns. If you have any further questions, we would be happy to receive your feedback.
>
> Thank you again for your support and valuable insights!
>
> Best wishes

---

> > ### Author Response · Authors · 2024-12-02
> > **Request for Reviewer [iRq4]'s Feedback**
> >
> > Dear Reviewer [iRq4],
> >
> > Thank you again for your time and effort in reviewing our paper.
> >
> > As the new Discussion Period ends on December 2nd at 24:00 AoE, we kindly remind you to review our rebuttal and shared responses.
> >
> > We have comprehensively responded to your concerns through detailed explanations and thorough experimental validation, and we also explain the contribution of our approach to real-time sleep monitoring. We, along with Reviewer [A2VX], sincerely believe that our paper has been strengthened a lot thanks to feedback.
> >
> > We are eagerly awaiting your feedback.
> >
> > Best wishes

---

### Official Review · Reviewer_Dg5B · 2024-11-03

**Soundness:** 4
**Presentation:** 3
**Contribution:** 3
**Rating:** 6
**Confidence:** 4

**Summary:**

This paper presents SleepSMC, a sleep stage classification framework that integrates uncertainty-based feature reweighting with modality-level contrastive learning to handle multimodal physiological data (EEG, EOG, EMG) effectively. The reweighting mechanism assigns weights to auxiliary modalities based on their uncertainty. Then, they use contrastive learning to align representations of the same sleep stage across different modalities. SleepSMC achieves modality-invariant embeddings that allow for robust performance even when only a single primary modality is available at test time. SleepSMC is evaluated on three public datasets, consistently outperforming SOTA baselines in both multimodal and unimodal settings.  They present ablation studies demonstrating that reweighting and contrastive learning are both effective individually and moreso when combined. Visualization analyses further demonstrate the model’s interpretability, showing clear, well-separated embeddings for each sleep stage.

**Strengths:**

This paper presents SleepSMC, a sleep stage classification framework that integrates uncertainty-based feature reweighting with modality-level contrastive learning to handle multimodal physiological data (EEG, EOG, EMG) effectively. The reweighting mechanism assigns weights to auxiliary modalities based on their uncertainty. Then, they use contrastive learning to align representations of the same sleep stage across different modalities. SleepSMC achieves modality-invariant embeddings that allow for robust performance even when only a single primary modality is available at test time. SleepSMC is evaluated on three public datasets, consistently outperforming SOTA baselines in both multimodal and unimodal settings.  They present ablation studies demonstrating that reweighting and contrastive learning are both effective individually and moreso when combined. Visualization analyses further demonstrate the model’s interpretability, showing clear, well-separated embeddings for each sleep stage.

Originality
The uncertainty reweighting technique is clever and novel.  Most importantly, it is both empirically and theoretically effective. The contrastive learning component, combined with reweighting, enables the model to create modality-invariant embeddings, allowing SleepSMC to generalize effectively across different primary modalities.


Quality
The quality of the work is high. The evaluation is thorough, providing empirical, theoretical, and qualitative support for their approach. Reweighting consistently outperformed non-reweighted setups in both multimodal and unimodal testing, with clear gains in accuracy, Macro F1, and Kappa scores, particularly under noisy conditions. Combining reweighting with contrastive learning further boosted performance compared to contrastive learning alone, showing that these methods work well together, especially with unreliable auxiliary modalities. SleepSMC also generalized effectively across target modalities, performing robustly whether EEG, EOG, or EMG was used as the primary modality, demonstrating flexibility across configurations.
In both multimodal and unimodal testing, SleepSMC maintained high accuracy, with unimodal performance benefiting notably from reweighting. t-SNE visualizations showed clear, well-separated clusters for each sleep stage, supporting the model’s modality-invariant embedding space and interpretability. These results confirm that SleepSMC’s reweighting and contrastive learning mechanisms enhance robustness and adaptability, making it a practical solution for real-world sleep staging.


Clarity
The work is mostly clearly presented. The methodology and results are well-structured, and the theoretical analysis is solid, providing strong support for the approach. However, the introduction of mathematical notation in Section 3 could be organized more effectively; starting with a table or list of symbols and definitions before diving into the equations would help clarify the notation and reduce cognitive load for the reader. The paper is otherwise transparent in discussing its limitations, and the overall clarity of the empirical findings is strong.


Significance
The work makes a meaningful contribution, adding to the literature on multimodal integration, particularly in scenarios where only unimodal data is available at test time—an approach that aligns well with many real-world applications.

**Weaknesses:**

1. The introduction of mathematical notation in Section 3 is somewhat disorganized and could be clarified better. Starting this section with a concise table or list of symbols and definitions would provide readers with a quick reference point, making it easier to follow the subsequent equations. This adjustment would enhance readability, especially for readers less familiar with the specific notation conventions used.

2. The work makes a solid contribution to the field, with gains in the range of 2-3%, which are statistically significant but don’t drastically improve performance over baseline methods. While these improvements are valuable they're somewhat incremental.  Perhaps the gains are more significant in other multimodal learning problems.

**Questions:**

An important aspect of multimodal classification models is cross-subject generalization. Given that real-world applications often involve new subjects with varying characteristics, an evaluation on unseen subjects would provide valuable insight into the model’s robustness. If cross-subject generalization was analyzed, could the authors include these results? Otherwise, adding this analysis would strengthen the paper’s practical relevance.

Additionally, while the reported improvements are statistically significant, they appear relatively modest, with gains in the range of 2-3%. Could the authors elaborate on why these incremental gains are meaningful in the context of sleep staging? Providing additional context on the impact of these gains for real-world applications would clarify the value of these results.

Although the reweighting technique is effective within sleep staging, it would be useful to understand its potential applicability beyond this domain. Do the authors see possible applications of uncertainty-based reweighting in other multimodal learning tasks? Expanding on this would highlight the versatility and broader impact of the proposed method.

In terms of practical applications, it would be helpful to know how this model performs in real-world scenarios where data quality and availability are less controlled. Could the model handle dynamic shifts in modality reliability, and do the authors envision practical deployments for health monitoring or similar applications?

---

> ### Author Response · Authors · 2024-11-20
> **Response to [Dg5B] (Part 1/n)**
>
> Thank you for your comprehensive recognition and support of our work! We understand that your main concerns are related to certain experimental details and writing aspects of the paper. We address them below:
>
> 1. **A concise list of symbols to better clarify notation?**
>
>    We have added such a list in Section A.1 of the revised appendix. Thank you for your suggestion, which has improved our paper!
>
> 2. **Solid contribution in sleep staging but without significant breakthroughs. Could it yield greater benefits in other multimodal learning problems?**
>
>    Achieving a 2-3% improvement in the sleep staging field is already a substantial contribution. Furthermore, our paper primarily focuses on enhancing sleep staging comfort in ubiquitous scenarios. However, our method does indeed have the potential to be applied in other fields. Thank you for pointing this out—we have included this possibility in the "Discussion and Limitation" section of the revised appendix.
>
> 3. **Cross-subject experiments**
>
>    As described in the experimental settings of Section 5.2, all our experiments adopt a cross-subject setup, considering that physiological signal models are typically designed to generalize in such scenarios. In all experiments, the training and testing sets are cross-subject, and we report aggregate results over five different splits. We apologize for any confusion and have highlighted the *cross-subject* setup in Section 5.2.
>
> 4. **Could the model handle dynamic shifts in modality reliability, and do the authors envision practical deployments for health monitoring or similar applications?**
>
>    Yes, the model can handle dynamic shifts in modality reliability. During training, the reliability of each modality is re-evaluated at every step, and relevant metrics are recalculated. Our method ultimately enhances the overall robustness of the model, which allows it to adapt to reliability changes during the testing phase to a certain extent. What a coincidence! We have indeed envisioned practical deployments for health monitoring. We are currently developing a wearable single-modality sleep monitoring device and plan to deploy this model in such applications.

---

> ### Author Response · Authors · 2024-11-27
> **Thanks for Reviewer [Dg5B]**
>
> Dear Reviewer,
>
> Thank you for taking the time and effort to review our paper, and thank you for your recognition of our paper.
>
> We have provided a comprehensive response to your concerns through detailed explanations.
>
> We hope we have addressed your concerns. If you have any further questions, we would be delighted to hear your feedback.
>
> Thank you again for your support and valuable insights!
>
> Best regards

---

> > ### Author Response · Authors · 2024-12-02
> > **Request for Reviewer [Dg5B]'s Feedback**
> >
> > Dear Reviewer [Dg5B],
> >
> > Thank you again for your time and effort in reviewing our paper.
> >
> > As the new Discussion Period ends on December 2nd at 24:00 AoE, we kindly remind you to review our rebuttal and shared responses.
> >
> > We have comprehensively responded to your concerns through detailed explanations and thorough experimental validations. We, along with Reviewer [A2VX], sincerely believe that our paper has been strengthened a lot thanks to feedback.
> >
> > We are eagerly awaiting your feedback.
> >
> > Best wishes

---

### Author Response · Authors · 2024-11-20
**Summary of revision**

Dear reviewers, AC, SAC, and PC,

We sincerely thank you for your time and suggestions. As a summary, we are grateful that merits in novelty, methodology, theory, and empirical results are favored by reviewers:

- **Reviewer Dg5B**: find our work *'clever and novel'*, *'empirically and theoretically effective'*, *'high quality'*, *'clearly presented'*, and *'meaningful contribution'*
- **Reviewer iRq4**: *'good organization'*
- **Reviewer HHyg**: *'certain practical significance'*
- **Reviewer A2VX**: *'well-structured and clear'*, *'easy to follow'*, and *'extensive experiments'*

We revised the paper according to the reviewers' suggestions. The newer version of the paper is uploaded (The modified position is highlighted in blue).

Based on the reviewers' suggestions, we have supplemented and revised the manuscript in the following key areas:

- **Experiments**: We have added four state-of-the-art methods to the experiments across three datasets under two scenario settings (including a multimodal testing scenario and three unimodal testing scenarios). These methods encompass two sequence-to-sequence sleep staging approaches, one multimodal robustness approach, and one multimodal contrastive coordination method. Additionally, we have supplemented ablation results using different weighted metrics and analyzed the computational complexity of our method. It is worth noting that all our original experiments were conducted under a cross-subject setting, which holds significant relevance for physiological signal analysis.

- **Writing**: We have reorganized and supplemented the Theorems and Proofs in the appendix and enlarged the text in Figures 3 and 4 for better readability. We also corrected and checked a grammatical error, following the suggestion from reviewer 'HHyg.' Furthermore, we added a list of symbols in the appendix, as recommended by reviewer 'Dg5B.' We deeply appreciate these valuable comments.

Thank you for your help in making this work stronger.

Although most reviewers provided positive feedback, we would still like to emphasize the following points:

- **Motivation and Significance**: Multimodal monitoring devices are often difficult to wear, uncomfortable during the monitoring process, and lack robustness. Our work aims to reduce the number of modalities required during the testing phase through multimodal coordination learning while ensuring comfort and improving accuracy. Our initial motivation is to achieve comfortable, simple, and highly accurate sleep monitoring in ubiquitous scenarios, such as home environments.

- **Innovation**: Our work is not only innovative at the application level but also demonstrates sufficient methodological novelty. Specifically, our method incorporates supervised modality-level instance contrastive coordination and uncertainty-based weighting. The former introduces label information into the alignment of multiple modality relationships to improve accuracy, while the latter dynamically filters information during transmission to enhance robustness.

We sincerely our response can address all you concerns:) If you have any questions, please let us know:)

---

### Author Response · Authors · 2024-11-24
**Dear Reviewers: Request for your Feedback on Our Rebuttal Responses**

Dear Reviewers,

Thank you for your time and effort in reviewing our paper. We truly appreciate the positive feedback from the reviewers, recognizing our work as novel (Reviewer Dg5B), well-structured and clear (Reviewers A2VX and iRq4), and of practical significance (Reviewers HHyg and Dg5B).

As the discussion period ends on Nov 26 at 24:00 AoE, we kindly remind you to review our rebuttal and shared responses.

We have provided a comprehensive response to your concerns through detailed explanations and thorough experimental validations. We are eagerly awaiting your feedback.

Thank you again for your support and valuable insights!

Best regards

---

### Author Response · Authors · 2024-12-03

Dear **Reviewer Dg5B** and **Reviewer iRq4**,

We would like to express our heartfelt gratitude for the recognition and support you have shown us in the initial stage of the review process. We are also deeply thankful for your valuable suggestions, which have truly enhanced the quality of our revised paper.

We have made every effort to carefully address and resolve the concerns you raised, and we hope that our revisions meet your expectations. As the discussion period is coming to a close, we humbly and earnestly hope to earn your continued strong support. If there are any further questions or concerns, we would be more than willing to address them.

Dear **Reviewer A2VX**,

We are truly delighted that we have been able to address all of your concerns, and we are deeply grateful for your decision to raise our score. Your support and encouragement mean a great deal to us.

As the discussion period is nearing its end, we sincerely welcome any further questions or feedback you may have. Your insights have been invaluable to us, and we would be more than happy to address any additional concerns.

---

### Meta-Review · Area_Chair_HAdZ · 2024-12-12

**Metareview:**

The paper introduces SleepSMC, a novel framework for ubiquitous sleep staging that addresses the challenges of leveraging multimodal data for training while ensuring high performance with unimodal data during testing. SleepSMC employs supervised modality-level instance contrastive coordination to align category-related features across modalities and utilizes uncertainty-based feature weighting to prioritize reliable auxiliary modalities during training. Tested on three public datasets (ISRUC-S3, MASS-SS3, Sleep-EDF-78), SleepSMC achieves state-of-the-art performance in both multimodal and unimodal testing scenarios, bridging the gap between the complex multimodal settings of clinical sleep staging and practical, real-world applications. This approach enhances robustness, interpretability, and real-world applicability for sleep monitoring in ubiquitous scenarios. The initial weakness of this paper mainly comes from the technical novelty and the evaluation.

In the rebuttal, the authors provides more clarifications and has supplement more experiments that demonstrate the effectiveness. 3 reviewers give accept and 1 reviewer still gives "lower than borderline" due to a confusing point in the experiment part. I concisely check the discussions between authors and the reviewer HHyg  and believe that the supplementary results follow a reasonable experiment design. However, the previous results in the original paper should also be included and the gap should be explained. Otherwise, it is confusing why previous methods produce a higher baseline. At least, the authors should compare it using the original setting. Overall, the method is tailored in sleep staging application and each module is well justified. Since I believe most concerns have been addressed, I still vote towards a borderline accept.

**Additional Comments On Reviewer Discussion:**

In the rebuttal, the authors provides more clarifications and has supplement more experiments that demonstrate the effectiveness, addressing most of the concerns in clarification and experiments. 3 reviewers give accept and 1 reviewer still gives "lower than borderline" due to a confusing point in the experiment part. I concisely check the discussions between authors and the reviewer HHyg. The issue is that the authors conduct the experiments on S3 following a new experiment protocol. After reading the original baseline paper, I believe that the supplementary results by authors follow a reasonable experiment design. However, the previous results in the original paper should also be included or compared and the gap should be explained. Otherwise, it is confusing why previous methods produce a higher baseline. At least, the authors should compare it using the original setting.

---

### Decision · Program_Chairs · 2025-01-22

Accept (Poster)